# Permute-and-Flip: A new mechanism for differentially private selection

**Ryan McKenna and Daniel Sheldon**
College of Information and Computer Sciences
University of Massachusetts, Amherst
Amherst, MA 01002
{ rmckenna, sheldon }@cs.umass.edu

## Abstract

We consider the problem of *differentially private selection*. Given a finite set of candidate items and a quality score for each item, our goal is to design a differentially private mechanism that returns an item with a score that is as high as possible. The most commonly used mechanism for this task is the exponential mechanism. In this work, we propose a new mechanism for this task based on a careful analysis of the privacy constraints. The expected score of our mechanism is always at least as large as the exponential mechanism, and can offer improvements up to a factor of two. Our mechanism is simple to implement and runs in linear time.

## 1 Introduction

The exponential mechanism [28] is one of the most fundamental mechanisms for differential privacy. It addresses the important problem of *differentially private selection*, or selecting an item from a set of candidates that approximately maximizes some objective function. The exponential mechanism was introduced soon after differential privacy itself, and has remained the dominant mechanism for private selection since.

The exponential mechanism is simple, easy to implement, runs in linear time, has good theoretical and practical performance, and solves an important problem. It can be used directly as a competitive mechanism for computing simple statistics like medians or modes [14, 16, 29]. Furthermore, it is an integral part of several more complex differentially private mechanisms for a range of tasks, including linear query answering [19], heavy hitter estimation [27], synthetic data generation [13, 36], dimensionality reduction [5, 12, 23], linear regression [2, 34], and empirical risk minimization [4, 9, 24, 32].

In this work, we propose the *permute-and-flip mechanism* as an alternative to the exponential mechanism for the task of differentially private selection. It enjoys the same desirable properties of the exponential mechanism stated above, and its expected error is never higher, but can be up to two times lower than that of the exponential mechanism. Furthermore, we show that in reasonable settings no better mechanism exists: the permute-and-flip mechanism is Pareto optimal, and, if $\epsilon \geq \log(\frac{1}{2}(3 + \sqrt{5})) \approx 0.96$, is optimal in a reasonable sense "overall".

The permute-and-flip mechanism serves as a drop-in replacement for the exponential mechanism in existing and future mechanisms, and immediately offers utility improvements. The utility improvements of up to $2\times$ over the state-of-the-art will impact practical deployments of differential privacy, where choosing the right privacy-utility trade-off is already a challenging social choice [1].

## 2 Preliminaries

### 2.1 Differential Privacy

A dataset $D$ is a collection of individual data coming from the universe of all possible datasets $\mathcal{D}$. We say datasets $D$ and $D'$ are neighbors, denoted $D \sim D'$, if they differ in the data of a single individual.

Differential privacy is a mathematical privacy definition, and a property of a mechanism, that guarantees the output of the mechanism will not differ significantly (in a probabilistic sense) between any two neighboring datasets.

**Definition 1** (Differential Privacy). *A randomized mechanism $\mathcal{M} : \mathcal{D} \rightarrow \mathcal{R}$ is $\epsilon$-differentially private, if and only if:*
$$\Pr[\mathcal{M}(D) \in S] \leq \exp(\epsilon) \Pr[\mathcal{M}(D') \in S]$$
*for all neighboring datasets $D \sim D'$ and all possible subsets of outcomes $S \subseteq \mathcal{R}$.*

The sensitivity of a function is an important quantity to consider when designing differentially private mechanisms, which measures how much a function can change between two neighboring datasets.

**Definition 2** (Sensitivity). *The sensitivity of a real-valued function $q : \mathcal{D} \rightarrow \mathbb{R}$ is defined to be:*
$$\Delta_q = \max_{D \sim D'} |q(D) - q(D')|.$$

### 2.2 Private Selection

In this work, we study the problem of *differentially private selection*. Given a finite set of candidates $\mathcal{R} = \{1, \ldots, n\}$ and an associated quality score function $q_r : \mathcal{D} \rightarrow \mathbb{R}$ for each $r \in \mathcal{R}$, our goal is to design a differentially private mechanism $\mathcal{M}$ that returns a candidate $r$ that approximately maximizes $q_r(D)$. The function $q_r(D)$ is typically a measure of how well the candidate $r$ captures some statistic or property of the dataset $D$. A simple example is the most common medical condition, where $q_r$ counts the number of individuals with medical condition $r$ [16].

The only assumption we make is that the sensitivity of $q_r$ is bounded above by $\Delta$ for each $r \in \mathcal{R}$. We consider mechanisms $\mathcal{M}$ that only depend on the dataset $D$ through the quality scores $q_r(D)$. Thus, for notational convenience, we drop the dependence on $D$, and treat a mechanism as a function of the quality scores instead. Specifically, we use $q_r \in \mathbb{R}$ to denote a quality score, $\vec{q} = [q_r]_{r \in \mathcal{R}}$ to denote the vector of quality scores, and $\mathcal{M}(\vec{q})$ to denote a mechanism executed on the quality score vector $\vec{q}$. We define a notion of regularity, describing properties we would like in a mechanism for this task:

**Definition 3** (Regularity). *A mechanism $\mathcal{M} : \mathbb{R}^n \rightarrow \mathcal{R}$ is regular if the following holds:*

*Symmetry: For any permutation $\pi : \mathcal{R} \rightarrow \mathcal{R}$ and associated permutation matrix $\Pi \in \mathbb{R}^{n \times n}$,*
$$\Pr[\mathcal{M}(\vec{q}) = r] = \Pr[\mathcal{M}(\Pi \vec{q}) = \pi(r)]. \tag{1}$$

*Shift-invariance: For all constants $c \in \mathbb{R}$,*
$$\Pr[\mathcal{M}(\vec{q}) = r] = \Pr[\mathcal{M}(\vec{q} + c\vec{1}) = r]. \tag{2}$$

*Monotonicity: If $q_r \leq q_r'$ and $q_s \geq q_s'$ for all $s \neq r$, then*
$$\Pr[\mathcal{M}(\vec{q}) = r] \leq \Pr[\mathcal{M}(\vec{q}') = r]. \tag{3}$$

Informally, a symmetric mechanism is one where the quality scores can be permuted arbitrarily without affecting the distribution of outcomes. This avoids pathologies where a mechanism can appear to do well for a particular quality score vector, but only because it has a built-in bias towards certain outcomes. Similarly, a shift-invariant mechanism is one where a constant can be added to all quality scores without changing the distribution of outcomes. A mechanism is monotonic if increasing one quality score while decreasing others will increase the probability on that outcome, and decrease the probability on all other outcomes. A mechanism that satisfies all of these criteria is called *regular*.

Mechanisms that are not regular have undesirable pathologies. Thus, we restrict our attention to regular mechanisms in this work. Beyond regularity, the main criteria we use to evaluate a mechanism is the error random variable:
$$\mathcal{E}(\mathcal{M}, \vec{q}) = q_* - q_{\mathcal{M}(\vec{q})} \tag{4}$$

where $q_* = \max_{r \in \mathcal{R}} q_r$ is the optimal quality score.

---
**Algorithm 1:** Permute-and-Flip Mechanism, $\mathcal{M}_{PF}(\vec{q})$
---
$q_* = \max_r q_r$
**for** $r$ *in RandomPermutation($\mathcal{R}$)* **do**

> $p_r = \exp\left(\frac{\epsilon}{2\Delta}(q_r - q_*)\right)$
> **if** *Bernoulli($p_r$)* **then**
> > **return** $r$
>
> **end**

**end**
---

## 2.3  Exponential Mechanism

The exponential mechanism is a mechanism that is both classical and state-of-the-art for the task of differentially private selection.

**Definition 4** (Exponential mechanism). *Given a quality score vector $\vec{q} \in \mathbb{R}^n$, the exponential mechanism is defined by:*

$$\Pr[\mathcal{M}_{EM}(\vec{q}) = r] \propto \exp\left(\frac{\epsilon}{2\Delta} q_r\right).$$

It is well-known that the exponential mechanism is $\epsilon$-differentially private, and it is easy to show that it also satisfies the regularity conditions in Definition 3. In addition, it is possible to bound the error of the exponential mechanism, both in expectation and in probability:

**Proposition 1** (Utility Guarantee of $\mathcal{M}_{EM}$ [8, 16]). *For all $\vec{q} \in \mathbb{R}^n$ and all $t \geq 0$,*

$$\mathbb{E}[\mathcal{E}(\mathcal{M}_{EM}, \vec{q})] \leq \frac{2\Delta}{\epsilon}\log(n), \qquad \Pr[\mathcal{E}(\mathcal{M}_{EM}, \vec{q}) \geq \frac{2\Delta}{\epsilon}(\log(n) + t)] \leq \exp(-t).$$

# 3  Permute-and-Flip Mechanism

In this section, we propose a new mechanism, $\mathcal{M}_{PF}$, which we call the "permute-and-flip" mechanism. Just like the exponential mechanism, it is simple, easy to implement, and runs in linear time. It is stated formally in Algorithm 1. The mechanism works by iterating through the set of candidates $\mathcal{R}$ in a random order. For each item, it flips a biased coin, and returns that item if the coin comes up heads. The probability of heads is an exponential function of the quality score, which encourages the mechanism to return results with higher quality scores. The mechanism is guaranteed to terminate with a result because if $q_r = q_*$, then the probability of heads is 1.

**Theorem 1.** *The Permute-and-Flip mechanism $\mathcal{M}_{PF}$ is regular and $\epsilon$-differentially private.*

Proofs of all results appear in the supplement; in addition, the main text will contain some proof sketches. The proof of this theorem uses Proposition 2 (below) and a direct analysis of the probability mass function of $\mathcal{M}_{PF}$. Note that the condition in Proposition 2 can be immediately verified to hold (with equality) when $q_r \leq q_* - 2\Delta$ by observing that $p_r$ in Algorithm 1 changes by exactly $\exp(\epsilon)$ when $q_r$ increases by $2\Delta$, and by a short argument conditioning on the random permutation. The proof using the pmf also handles the case when increasing $q_r$ by $2\Delta$ causes item $r$ to have maximum score.

## 3.1  Derivation

We now describe the principles underlying the permute-and-flip mechanism and intuition behind its derivation. To define a mechanism, we must specify the value of $\Pr[\mathcal{M}(\vec{q}) = r]$ for every $(\vec{q}, r)$ pair. Intuitively, we would like to place as much probability mass as possible on the items with the highest score, and as little mass as possible on other items, subject to the constraints of differential privacy. For regular mechanisms, these constraints simplify greatly:

**Proposition 2.** *A regular mechanism $\mathcal{M} : \mathbb{R}^n \to \mathcal{R}$ is $\epsilon$-differentially private if:*

$$\Pr[\mathcal{M}(\vec{q}) = r] \geq \exp(-\epsilon)\Pr[\mathcal{M}(\vec{q} + 2\Delta\vec{e}_r) = r]$$

*for all $(\vec{q}, r)$, where $\vec{e}_r$ is the unit vector with a one at position $r$.*

The proof (in the supplement) argues that it is only necessary to compare $\vec{q}$ to the quality-score vector with $q'_r = q_r + \Delta$ and $q'_s = q_s - \Delta$ for all $s \neq r$, which, by monotonicity, is the *worst-case* neighbor of $\vec{q}$. By shift-invariance, the mechanism is identical when $q'_r = q_r + 2\Delta$ and $q'_s = q_s$, or $\vec{q}' = \vec{q} + 2\Delta \vec{e}_r$, which leads to the constraint in the theorem.

This theorem allows allows us to reason about only one constraint for every $(\vec{q}, r)$ pair, instead of infinitely many. Ideally we would like to distribute probability to items as *un*evenly as possible, which would make these constraints tight (satisfied with equality). However, we can see by examining the overall numbers of constraints and variables that we cannot make all of them tight. For each score vector $\vec{q}$, there are: (1) $n = |\mathcal{R}|$ free variables (the probabilities of the mechanism run on $\vec{q}$), (2) $n$ inequality constraints (Proposition 2), and (3) one additional constraint that the probabilities sum to one. This is a total of $n$ inequality constraints and one equality constraint per $n$ variables. On average, we expect at most $n - 1$ of the inequality constraints to be tight, leading to $n$ linear constraints that are satisfied with equality for each group of $n$ variables.

The following recurrence for $\Pr[\mathcal{M}(\vec{q}) = r]$ defines a mechanism by selecting *certain* constraints to be satisfied with equality:

$$\Pr[\mathcal{M}(\vec{q}) = r] = \begin{cases} \exp\left(-\epsilon\right) \Pr[\mathcal{M}(\vec{q} + 2\Delta \vec{e}_r) = r] & q_r \leq q_* - 2\Delta \\ \frac{1}{n_*}\left(1 - \sum_{s:q_s < q_*} \Pr[\mathcal{M}(\vec{q}) = s]\right) & q_r = q_* \end{cases} \tag{5}$$

The privacy constraint is tight whenever $q_r \leq q_* - 2\Delta$ (Case 1). When $q_r$ is one of the maximum scores (Case 2, $q_r = q_*$), the sum-to-one constraint is used instead, in conjunction with symmetry; here, $n_*$ is the number of quality scores equal to $q_*$.

This recurrence defines a *unique* mechanism for quality score vectors on the $2\Delta$-lattice, i.e., for $\vec{q} \in \mathbb{R}^n_{2\Delta} = \{2\Delta \vec{s} : \vec{s} \in \mathbb{Z}^n\}$. To see this, note that the base case occurs when $n_* = n$, i.e., all scores are equal to the maximum and each item has probability $\frac{1}{n}$. Now consider an arbitrary $\vec{q} \in \mathbb{R}^n_{2\Delta}$ with $n_* = k$ maximum elements. By Case 2, the mechanism is fully defined by the probabilities assigned to items with non-maximum scores. By Case 1, the probability of selecting an item $r$ that does not have maximum score is defined by the probability of selecting $r$ with the score vector $\vec{q}' = \vec{q} + 2\Delta \vec{e}_r$; this also belongs to the $2\Delta$-lattice, and $q'_r > q_r$. Eventually, a score vector with $n_* = k+1$ maximum elements will be reached, moving closer to the base case of $n_* = n$.

The following recurrence generalizes the original and is well-defined for all $\vec{q} \in \mathbb{R}^n$, obtained by simply interpolating between the points in the $2\Delta$-lattice.

$$\Pr[\mathcal{M}(\vec{q}) = r] = \begin{cases} \exp\left(\frac{\epsilon}{2\Delta}(q_r - q_*)\right) \Pr[\mathcal{M}(\vec{q} + (q_* - q_r)\vec{e}_r) = r] & q_r < q_* \\ \frac{1}{n_*}\left(1 - \sum_{s:q_s < q_*} \Pr[\mathcal{M}(\vec{q}) = s]\right) & q_r = q_* \end{cases} \tag{6}$$

The only difference is Case 1, which is obtained by unrolling Case 1 of the original recurrence $(q_* - q_r)/2\Delta$ times so that the $r$th score becomes exactly $q_*$. The advantage is that the new expression is well defined for vectors that are not on the $2\Delta$-lattice.

Equation (6) defines a mechanism. In principle, it also gives a way to compute the probabilities of the mechanism for any fixed $\vec{q}$. The most direct approach to calculate these probabilities uses dynamic programming and takes exponential time. A smarter algorithm based on an analytic expression for the solution to the recurrence runs in $O(n^2)$ time (Appendix E), but is still unacceptably slow compared to the linear time exponential mechanism. Remarkably, it is not necessary to explicitly compute the probabilities of this mechanism, as the permute-and-flip mechanism solves this recurrence relation. As a result, we can simply run the simple linear-time Algorithm 1 and avoid computing the mechanism probabilities directly.

**Proposition 3.** $\mathcal{M}_{PF}$ *solves the recurrence relation in Equation* (6)*.*

*Proof (Sketch).* Case 2 is satisfied because $\mathcal{M}_{PF}$ is symmetric and a valid probability distribution. For Case 1, let $\vec{q}' = \vec{q} + (q_* - q_r)\vec{e}_r$ and consider applying $\mathcal{M}_{PF}$ to both $\vec{q}$ and $\vec{q}'$. In each case, the coin-flip probabilities are the same for all items *except* $r$, and the probability of selecting any given permutation is the same. The ratio $\Pr[\mathcal{M}_{PF}(\vec{q}) = r] / \Pr[\mathcal{M}_{PF}(\vec{q}') = r]$ can be shown to be *exactly* $p_r/p'_r$, where $p_r = \exp\left(\frac{\epsilon}{2\Delta}(q_r - q_*)\right)$ is the coin-flip probability with $\vec{q}$ and $p'_r = 1$ is the coin-flip probability with $\vec{q}'$. The ratio is exactly $\exp\left(\frac{\epsilon}{2\Delta}(q_r - q_*)\right)$, as required by Case 1. $\qquad \square$

| **Algorithm 2:** $\mathcal{M}_{EM}(\vec{q})$ | **Algorithm 3:** $\mathcal{M}_{PF}(\vec{q})$ |
|---|---|
| $q_* = \max_r q_r$ | $q_* = \max_r q_r$ |
| **repeat** | **repeat** |
| $\quad r \sim \text{Uniform}[\mathcal{R}]$ | $\quad r \sim \text{Uniform}[\mathcal{R}]$ |
| $\quad p_r = \exp\left(\frac{\epsilon}{2\Delta}(q_r - q_*)\right)$ | $\quad p_r = \exp\left(\frac{\epsilon}{2\Delta}(q_r - q_*)\right)$ |
| | $\quad \mathcal{R} = \mathcal{R} \setminus \{r\}$ |
| **until** *Bernoulli*$(p_r)$; | **until** *Bernoulli*$(p_r)$; |
| **return** $r$ | **return** $r$ |

## 4 Comparison with Exponential Mechanism

In this section, we compare the permute-and-flip and exponential mechanisms, both algorithmically and in terms of the error each incurs. One (unconventional) way to sample from the exponential mechanism is stated in Algorithm 2. This is a rejection sampling algorithm: an item is repeatedly sampled uniformly at random from the set $\mathcal{R}$ *with replacement* and returned with probability $p_r = \exp\left(\frac{\epsilon}{2\Delta}(q_r - q_*)\right)$. For the permute-and-flip mechanism, an item is repeatedly sampled uniformly at random from the set $\mathcal{R}$ *without replacement* and returned with the same probability. Sampling without replacement is mathematically equivalent to iterating through a random permutation, and hence Algorithm 3 is equivalent to Algorithm 1. These implementations are not recommended in practice, but are useful to illustrate connections between the two mechanisms.

Intuitively, sampling without replacement is better, because items that are not selected, which are likely to have low scores, are eliminated from future consideration. In fact, in Theorem 2 we prove that the permute-and-flip mechanism is never worse than the exponential mechanism in a very strong sense. Specifically, we show that the expected error of permute-and-flip is never larger than the exponential mechanism, and the probability of the error random variable exceeding $t$ is never larger for permute-and-flip (for any $t$). This is a form of *stochastic dominance* [18], and suggests it is always preferable to use permute-and-flip over the exponential mechanism, no matter what the risk profile is.

**Theorem 2.** $\mathcal{M}_{PF}$ *is never worse than* $\mathcal{M}_{EM}$. *That is, for all* $\vec{q} \in \mathbb{R}^n$ *and all* $t \geq 0$,

$$\mathbb{E}[\mathcal{E}(\mathcal{M}_{PF}, \vec{q})] \leq \mathbb{E}[\mathcal{E}(\mathcal{M}_{EM}, \vec{q})], \qquad \Pr[\mathcal{E}(\mathcal{M}_{PF}, \vec{q}) \geq t] \leq \Pr[\mathcal{E}(\mathcal{M}_{EM}, \vec{q}) \geq t]$$

As a direct consequence of Theorem 2, the permute-and-flip mechanism inherits the theoretical guarantees of the exponential mechanism (Proposition 1).

**Corollary 1.** *For all* $\vec{q} \in \mathbb{R}^n$ *and all* $t \geq 0$,

$$\mathbb{E}[\mathcal{E}(\mathcal{M}_{PF}, \vec{q})] \leq \frac{2\Delta}{\epsilon} \log(n), \qquad \Pr[\mathcal{E}(\mathcal{M}_{PF}, \vec{q}) \geq \frac{2\Delta}{\epsilon}(\log(n) + t)] \leq \exp(-t).$$

### 4.1 Analysis of Worst-Case Error

To further compare the two mechanisms, it is instructive to compare their expected errors for a particular class of score vectors. In particular, we examine score vectors that are *worst cases* for both mechanisms. This analysis will reveal that permute-and-flip can be up to $2\times$ better than the exponential mechanism, and that the upper bounds on expected error in Proposition 1 and Corollary 1 are within a factor of four of being tight.

**Proposition 4.** *The worst-case expected errors for both* $\mathcal{M}_{EM}$ *and* $\mathcal{M}_{PF}$ *occur when* $\vec{q} = (c, \ldots, c, 0) \in \mathbb{R}^n$ *for some* $c \leq 0$. *Let* $p = \exp\left(\frac{\epsilon}{2\Delta}c\right)$. *The expected errors for score vectors of this form are:*

$$\mathbb{E}[\mathcal{E}(\mathcal{M}_{EM}, \vec{q})] = \frac{2\Delta}{\epsilon} \log\left(\frac{1}{p}\right)\left[1 - \frac{1}{1 + (n-1)p}\right], \tag{7}$$

$$\mathbb{E}[\mathcal{E}(\mathcal{M}_{PF}, \vec{q})] = \frac{2\Delta}{\epsilon} \log\left(\frac{1}{p}\right)\left[1 - \frac{1 - (1-p)^n}{np}\right]. \tag{8}$$

*The worst-case expected errors are found by maximizing Equations* (7) *and* (8) *over* $p \in (0, 1]$.

Figure 1a shows the expected error of both mechanisms using Equations (7) and (8) for $n = 3$ and $p \in (0, 1]$. The error of $\mathcal{M}_{PF}$ is always lower than that of $\mathcal{M}_{EM}$, as expected by Theorem 2.

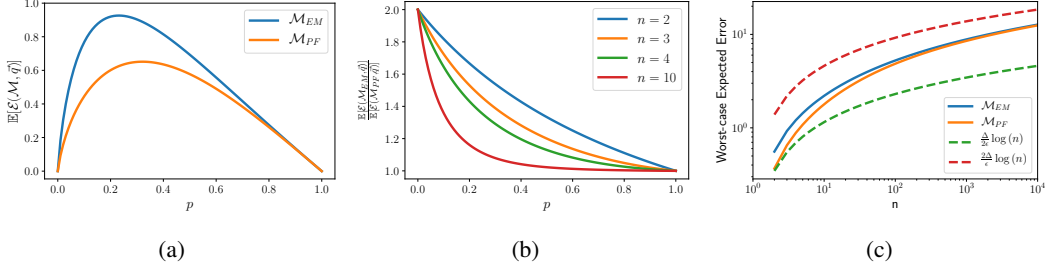

Figure 1: (a) Expected error of $\mathcal{M}_{EM}$ and $\mathcal{M}_{PF}$ as a function of $p$ for $n = 3$, $\epsilon = 1$, and $\Delta = 1$. (b) Ratio of expected errors between $\mathcal{M}_{EM}$ and $\mathcal{M}_{PF}$ as a function of $p$ for varying $n$. (c) Worst-case expected error of $\mathcal{M}_{EM}$ and $\mathcal{M}_{PF}$ as a function of $n$.

At the two extremes ($p = 0$ and $p = 1$), the expected error of both mechanisms is exactly 0, because: (1) when $p = 1$, all scores are equal to the maximum, and (2) when $p \to 0$, the total probability assigned to items with non-maximum scores vanishes. The maximum error for each mechanism occurs somewhere in the middle, typically near $p = \frac{1}{n}$. In fact, by substituting $p = \frac{1}{n}$ into Equation (8), we obtain:

**Proposition 5.** *For $\vec{q} = (c, \ldots, c, 0) \in \mathbb{R}^n$ with $c = -\frac{2\Delta}{\epsilon} \log n$, the expected error $\mathbb{E}[\mathcal{E}(\mathcal{M}_{PF}, \vec{q})]$ of permute-and-flip is at least $\frac{\Delta}{2\epsilon} \log(n)$. This implies that $\mathbb{E}[\mathcal{E}(\mathcal{M}_{EM}, \vec{q})] \geq \frac{\Delta}{2\epsilon} \log(n)$ as well, and that the upper bounds of Proposition 1 and Corollary 1 are within a factor of four of being tight.*

Figure 1b shows the ratio $\frac{\mathbb{E}[\mathcal{E}(\mathcal{M}_{EM}, \vec{q})]}{\mathbb{E}[\mathcal{E}(\mathcal{M}_{PF}, \vec{q})]}$ of expected errors of the two mechanisms for different values of $n$ for $p \in (0, 1]$. The result is independent of particular choices of $\epsilon$ and $\Delta$, as the ratio only depends on $\epsilon$ and $\Delta$ through $p$. We observe that:

- The ratio is always between one and two, and approaches two in the limit at $p \to 0$ (larger $\epsilon$).
- The required $p$ to achieve a fixed ratio decreases with $n$, and the ratio converges to one for all $p > 0$ as $n$ goes to infinity. This behavior is well-explained by the algorithmic comparison earlier in this section: as $n$ goes to infinity, the probability of sampling the same low-scoring item multiple times becomes negligible, so sampling without replacement ($\mathcal{M}_{PF}$) becomes essentially identical to sampling with replacement ($\mathcal{M}_{EM}$).

These results are for a *particular* class of (worst-case) quality-score vectors and not necessarily indicative of what will happen in applications. In our experiments with real quality-score vectors (Section 6) we observe ratios close to two for the values of $\epsilon$ that provide reasonable utility. We have never observed a ratio greater than two for any $\vec{q}$, and it is an open question whether this is possible. In practice, we can and do realize significant improvements even for large $n$.

Figure 1c compares the worst-case expected errors of $\mathcal{M}_{EM}$ and $\mathcal{M}_{PF}$ as a function of $n$ by numerically maximizing over $p$ in Equations (7) and (8) for different values of $n$ and $\epsilon = \Delta = 1$. For reference, we also plot the analytic upper and lower bounds from Proposition 1, Corollary 1, and Proposition 5. The ratio of worst-case expected error between $\mathcal{M}_{EM}$ and $\mathcal{M}_{PF}$ is largest at $n = 2$, and it decays towards 1 as $n$ increases. Again, this is explained by the algorithmic similarities between the two mechanisms as $n \to \infty$.

## 5 Optimality of Permute-and-Flip

In the previous section, we showed that permute-and-flip is never worse than the exponential mechanism, and is sometimes better by up to a factor of two. In other words, it *Pareto dominates* the exponential mechanism. In Proposition 6 we show that permute-and-flip is in fact *Pareto optimal* on the $2\Delta$-lattice $\mathbb{R}^n_{2\Delta}$ (see Section 3.1) with respect to the expected error. That is, any regular mechanism that is better than permute-and-flip for some $\vec{q} \in \mathbb{R}^n_{2\Delta}$ must be worse for some other $\vec{q}' \in \mathbb{R}^n_{2\Delta}$.

**Proposition 6** (Pareto Optimality). *If $\mathbb{E}[\mathcal{E}(M_{PF}, \vec{q})] > \mathbb{E}[\mathcal{E}(\mathcal{M}, \vec{q})]$ for some regular mechanism $\mathcal{M}$ and some $\vec{q} \in \mathbb{R}^n_{2\Delta}$, then there exists $\vec{q}' \in \mathbb{R}^n_{2\Delta}$ such that $\mathbb{E}[\mathcal{E}(M_{PF}, \vec{q}')] < \mathbb{E}[\mathcal{E}(\mathcal{M}, \vec{q}')]$.*

Pareto optimality is a desirable property that differentiates permute-and-flip from the exponential mechanism. However, there are many Pareto optimal mechanisms, so we would like additional

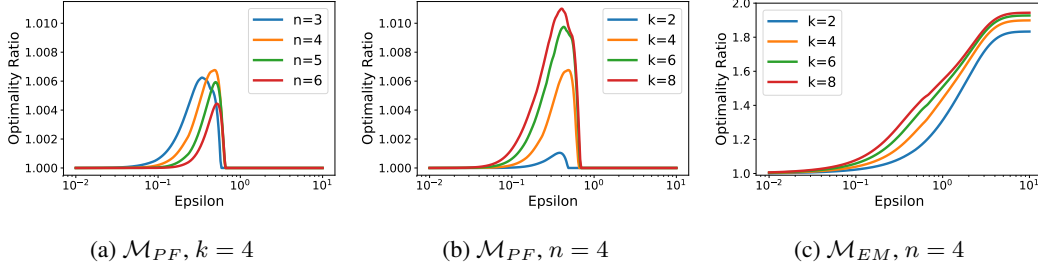

(a) $\mathcal{M}_{PF}$, $k = 4$          (b) $\mathcal{M}_{PF}$, $n = 4$          (c) $\mathcal{M}_{EM}$, $n = 4$

Figure 2: Optimality ratio for $\mathcal{M}_{PF}$ and $\mathcal{M}_{EM}$ for various $n$ and $k$.

assurance that permute-and-flip is in some sense the "right" one. To achieve this, we show that it is optimal in some reasonable "overall" sense. In particular, it minimizes the expected error *averaged* over a representative set of quality score vectors, as long as $\epsilon$ is sufficiently large.

**Theorem 3** (Overall Optimality). *For all regular mechanisms $\mathcal{M}$ and all $\epsilon \geq \log\left(\frac{1}{2}(3 + \sqrt{5})\right)$,*

$$\sum_{\vec{q} \in Q} \mathbb{E}[\mathcal{E}(\mathcal{M}_{PF}, \vec{q})] \leq \sum_{\vec{q} \in Q} \mathbb{E}[\mathcal{E}(\mathcal{M}, \vec{q})]$$

*where $Q = \{\vec{q} \in \mathbb{R}^n_{2\Delta} : q_* - q_r \leq 2\Delta k, q_* = 0\}$ for any integer constant $k \geq 0$.*

This theorem is proved by analyzing a linear program (LP) that describes the behavior of an optimal regular mechanism on the $2\Delta$-lattice, using the linear constraints described in Section 3.1 to enforce privacy and regularity, and the linear objective from the theorem. The result holds for the *bounded* lattice with $q_* = 0$ and all scores at most $k$ lattice points away from zero. Boundedness is required to have a finite number of variables and constraints. The restriction that $q_* = 0$ is without loss of generality: by shift-invariance, a regular mechanism is completely defined by its behavior on vectors with $q_* = 0$.

Theorem 3 *guarantees* that permute-and-flip is optimal if $\epsilon$ is large enough. For smaller $\epsilon$, we can empirically check how close to optimal $\mathcal{M}_{PF}$ is by solving the LP. This is computationally prohibitive in general, because the LP size grows quickly and becomes intractable for large $n$ and $k$, but we can make comparisons for smaller lattices. Figure 2 shows the "optimality ratio" of permute-and-flip and the exponential mechanism for various settings of $\epsilon$, $n$, and $k$. The optimality ratio of a mechanism $\mathcal{M}$ is the ratio $\frac{\sum_{\vec{q}} \mathbb{E}[\mathcal{E}(\mathcal{M}, \vec{q})]}{\sum_{\vec{q}} \mathbb{E}[\mathcal{E}(\mathcal{M}_*, \vec{q})]}$, where $\mathcal{M}_*$ is the optimal mechanism on the bounded $2\Delta$-lattice obtained by solving the linear program.

As shown in Figure 2a and Figure 2b, the optimality ratio for permute-and-flip is equal to one above the threshold, as expected. Furthermore, it barely exceeds one even when $\epsilon$ is below the threshold: the largest measured value is about $1.01$. The ratio grows slowly with $k$ (Figure 2b) and shows no strong dependence on $n$ (Figure 2a). For the exponential mechanism (Figure 2c), the optimality ratio is much more significantly larger than one, and generally increases with $\epsilon$, approaching two for larger $k$ and $\epsilon$. Interestingly, the optimality ratio approaches one for both mechanisms as $\epsilon \to 0$.

## 6 Experiments

We now perform an empirical analysis of the permute-and-flip mechanism. Our aim is to quantify the utility improvement from permute-and-flip relative to the exponential mechanism for different values of $\epsilon$ on real-world problem instances. We use five representative data sets from the DPBench study: HEPTH, ADULTFRANK, MEDCOST, SEARCHLOGS, and PATENT [20] and consider the tasks of mode and median selection. In each case, the candidates are the 1024 bins of a discretized domain. For each task, we construct the quality score vector and then *analytically* compute the expected error for a range of different $\epsilon$ for both the permute-and-flip and exponential mechanisms using their probability mass functions. Below we summarize our experimental findings; additional experimental results can be found in Appendix G.

**Mode.** For mode selection, the quality function is the number of items in the bin, which has sensitivity one. Figure 3a shows expected error as a function of $\epsilon$ for the HEPTH data set. Note that expected error is plotted on a log scale, while $\epsilon$ is plotted on a linear scale, and we truncate

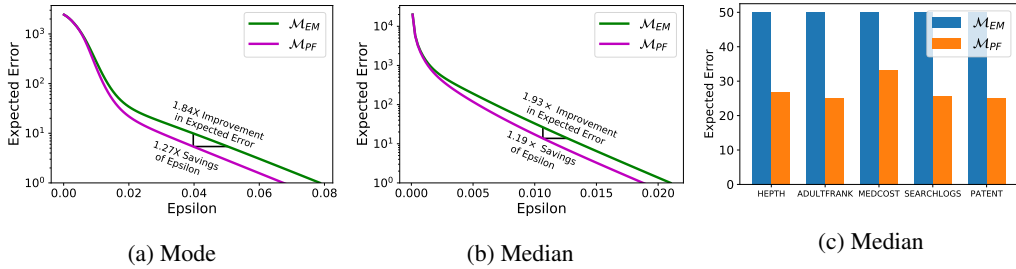

| (a) Mode | (b) Median | (c) Median |

Figure 3: (a) and (b) Expected error of $\mathcal{M}_{EM}$ and $\mathcal{M}_{PF}$ on the HEPTH dataset for varying $\epsilon$. (c) Expected error of $\mathcal{M}_{PF}$ on five datasets for $\epsilon$ where expected error of $\mathcal{M}_{EM}$ is 50.

the plot when the expected error falls below one. The ratio of the expected error of the exponential mechanism to that of permute-and-flip ranges from one (for smaller $\epsilon$) to two (for larger $\epsilon$). For the range of $\epsilon$ that provide reasonable utility, the improvement is closer to two. For example, at $\epsilon = 0.04$, the ratio is 1.84. The expected error of $\mathcal{M}_{PF}$ at this value of $\epsilon$ is about 5.4, and $\mathcal{M}_{EM}$ would need about 1.27 times larger privacy budget to achieve the same utility.

**Median.**  For median selection, the quality function is the (negated) number of individuals that must be added or removed to make a given bin become the median, which is also a sensitivity one function [29]. Figure 3b again shows the expected error as a function of $\epsilon$ for the HEPTH data set. Again, the ratio of expected errors ranges from one (for smaller $\epsilon$) to two (for larger $\epsilon$). For the range of $\epsilon$ that provide reasonable utility, the improvement is closer to two. For example, at $\epsilon = 0.01$, the ratio is 1.93. The expected error of $\mathcal{M}_{PF}$ at this value of $\epsilon$ is about 13.7, and $\mathcal{M}_{EM}$ would need about 1.19 times larger privacy budget to achieve the same utility.

In Figures 3a and 3b, the expected errors of $\mathcal{M}_{EM}$ and $\mathcal{M}_{PF}$ become *approximately parallel lines* as $\epsilon$ increases. Because the plots use linear scale for $\epsilon$ and logarithmic scale for expected error this means that the error of both mechanisms behaves approximately as $c \exp(-\epsilon)$ for some $c$. Additionally, $\mathcal{M}_{PF}$ offers an asymptotically constant *multiplicative* improvement in expected error (a factor of two) and an *additive* savings of $\epsilon$. For the range of $\epsilon$ that demonstrate the most reasonable privacy-utility tradeoffs, this additive improvement is a meaningful fraction of the privacy budget.

In Figure 3c we plot the expected error of $\mathcal{M}_{EM}$ and $\mathcal{M}_{PF}$ on all five data sets. For each dataset, we use the value of $\epsilon$ where $\mathcal{M}_{EM}$ gives a expected error of 50. This allows us to plot all datasets on the same scale for some $\epsilon$ that gives a reasonable tradeoff between privacy and utility. The improvements are significant, and close to a factor of two for all data sets.

## 7   Related Work

The exponential mechanism and the problem of differentially private selection have been studied extensively in prior work [3, 5, 6, 10, 11, 15, 21, 25, 26, 30, 31, 33, 35].

The most common alternative to the exponential mechanism for the private selection problem is *report noisy max* [16], which adds noise to each quality score and outputs the item with the largest noisy score. While we did not compare directly to this mechanism, our initial findings (Appendix F) indicate that it is competitive with the exponential mechanism, but neither mechanism Pareto dominates the other — report noisy max is better for some quality score vectors, while the exponential mechanism is better for others. Several other mechanisms have been proposed for the private selection problem that may work better under different assumptions and special cases [5, 10, 11, 25, 30, 35].

A generalization of the exponential mechanism was proposed in [31] that can effectively handle quality score functions with varying sensitivity. This technique works by defining a new quality score function that balances score and sensitivity and then running the exponential mechanism, and is therefore also compatible with the permute-and-flip-mechanism. The exponential mechanism was also studied in [15], where the focus was to improve the privacy analysis for a composition of multiple sequential executions of the exponential mechanism. They also show that the analysis can be improved in some cases by using a measure of the *range* of the score function instead of the sensitivity (though in commonly-used score functions the range and sensitivity usually coincide). This improvement is orthogonal to our approach, and it is straightforward to extend the analysis of the permute-and-flip mechanism in a similar way.

A new mechanism for *private selection from private candidates* was studied in [26]. Instead of assuming the quality functions have bounded sensitivity, it is assumed that the quality functions are themselves differentially private mechanisms. This relaxed assumption is appealing for many problems where the traditional exponential mechanism does not apply, like hyperparameter optimization.

The optimality of the exponential mechanism was studied in [3], where the authors considered linear programs for computing mechanisms that are optimal on average (similar to our Theorem 3). They restricted attention to scenarios where the input/output universe of the mechanism is a graph, and each node is associated with a database. They argued that the optimal mechanism should satisfy privacy constraints with equality for connected nodes in this graph, and showed that the exponential mechanism was optimal up to a constant factor of two in this setting.

Other works have carefully analyzed privacy constraints to construct optimal mechanisms for other tasks and privacy definitions, including predicate counting queries [17], information theoretic quantities [22], and generic low-sensitivity functions [7].

## 8 Conclusions and Open Questions

In this work we proposed permute-and-flip, a new mechanism for differentially private selection that can be seen as a replacement for the exponential mechanism. For every set of scores, the expected error of the permute-and-flip mechanism is not higher than the expected error of exponential mechanism, and can be lower by a factor of two; we observe factors close to two in real-world settings. Furthermore, we prove that the permute-and-flip mechanism is optimal in a fairly strong sense overall. Improving the exponential mechanism by a factor between one and two has the potential for wide-reaching impact, since it is one of the most important primitives in differential privacy.

Some remaining open questions are:

1. We focused primarily on the utility improvements offered by permute-and-flip in this work. In some cases, permute-and-flip may also offer runtime improvement. Specifically, if $q_*$ is known a-priori, then permute-and-flip can potentially terminate early without evaluating all $n$ quality scores. Identifying situations where this potential benefit can be realized and provide meaningful improvement is an interesting open question.

2. We demonstrated meaninful improvement over the exponential mechanism for simple tasks like median and mode estimation. It would be interesting to apply permute-and-flip to more advanced mechanisms that use the exponential mechanism, and quantify the improvement there.

3. Our overall optimality result restricts to score vectors on the bounded $2\Delta$-lattice. It would be interesting to understand more fully the nature of optimal mechanisms on more general domains or with other ways of averaging or aggregating over score vectors.

## Broader Impact

Our work fits in the established research area of differential privacy, which enables the positive societal benefits of gleaning insight and utility from data sets about people while offering formal guarantees of privacy to individuals who contribute data. While these benefits are largely positive, unintended harms could arise due to misapplication of differential privacy or misconceptions about its guarantees. Additionally, difficult social choices are faced when deciding how to balance privacy and utility. Our work addresses a foundational differential privacy task and enables better utility-privacy tradeoffs within this broader context.

## Acknowledgements

We would like to thank Gerome Miklau and the anonymous reviewers for their helpful comments to improve the paper. This work was supported by the National Science Foundation under grants CNS-1409143, IIS-1749854, IIS-1617533, and by DARPA and SPAWAR under contract N66001-15-C-4067.

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
