[Supplementary Material]

# A   Probability Mass Function of $\mathcal{M}_{PF}$

We begin by deriving two different expressions for the probability mass function of $\mathcal{M}_{PF}$, which we will reference in other proofs throughout the supplement.

**Lemma 1.** *The probability mass function (pmf) of $\mathcal{M}_{PF}$ can be expressed as:*

$$\Pr[\mathcal{M}_{PF}(\vec{q}) = r] = p_r \sum_{\pi} \frac{1}{n!} \prod_{s:\pi(s)<\pi(r)} (1 - p_s)$$

*where $\pi$ is a permutation and $p_r = \exp\left(\frac{\epsilon}{2\Delta}(q_r - q_*)\right)$.*

*Proof.* Let $X_s$ be the event that the $s$th coin is heads, and let $\pi$ be a random permutation. The events $X_s$ are independent. The $r$th item is selected if $X_r$ is true, and $X_s$ is false for all $s$ that come before $r$ in the permutation $\pi$, that is:

$$\begin{aligned}
\Pr[\mathcal{M}_{PF}(\vec{q}) = r] &= \Pr\left[X_r \cap \left(\bigcap_{s:\pi(s)<\pi(r)} \neg X_s\right)\right] \\
&= \frac{\Pr[X_r]}{n!} \sum_{\pi} \prod_{s:\pi(s)<\pi(r)} (1 - \Pr[X_s]) \\
&= \frac{p_r}{n!} \sum_{\pi} \prod_{s:\pi(s)<\pi(r)} (1 - p_s).
\end{aligned}$$

$\square$

**Lemma 2.** *An equivalent expression for the probability mass function of $\mathcal{M}_{PF}$ is:*

$$Pr[\mathcal{M}_{PF}(\vec{q}) = r] = p_r \sum_{\substack{S \subseteq \mathcal{R} \\ r \notin S}} \frac{(-1)^{|S|}}{|S|+1} \prod_{s \in S} p_s.$$

*Proof.* Let $X_s$ again denote the event that the $s$th coin is heads. Let $\pi$ be a random permutation and let $Y_s$ be the event $X_s \cap (\pi(s) < \pi(r))$, or "the $s$th coin is heads and appears before the $r$th coin in the random permuation". Note that the events $X_r$ and $Y_s$ are independent for $r \neq s$.

By independence and the inclusion-exclusion principle:

$$\begin{aligned}
\Pr[\mathcal{M}_{PF}(\vec{q}) = r] &= \Pr\left[X_r \cap \left(\neg \bigcup_{s \neq r} Y_s\right)\right] \\
&= \Pr[X_r]\left(1 - \Pr\left[\bigcup_{s \neq r} Y_s\right]\right) \\
&= \Pr[X_r]\left(1 - \sum_{\substack{S \subseteq \mathcal{R} \\ r \notin S \\ |S| \geq 1}} (-1)^{|S|} \Pr\left[\bigcap_{s \in S} Y_s\right]\right)
\end{aligned}$$

We now split the event $\bigcap_{s \in S} Y_S$, or "all coins in $S$ appear before $r$ and are heads", into the conjunction of the events "all coins in $S$ appear before $r$" and "all coins in $S$ are heads", and continue as:

$$\Pr[\mathcal{M}_{PF}(\vec{q}) = r] = \Pr[X_r]\left(1 - \sum_{\substack{S \subseteq \mathcal{R} \\ r \notin S \\ |S| \geq 1}} (-1)^{|S|} \cdot \Pr\left[\bigcap_{s \in S} (\pi(s) < \pi(r))\right] \cdot \Pr\left[\bigcap_{s \in S} X_s\right]\right)$$

$$= \Pr[X_r]\left(1 - \sum_{\substack{S \subseteq \mathcal{R} \\ r \notin S \\ |S| \geq 1}} (-1)^{|S|} \frac{1}{|S|+1} \prod_{s \in S} \Pr[X_s]\right)$$

$$= \Pr[X_r] \sum_{\substack{S \subseteq \mathcal{R} \\ r \notin S}} \frac{(-1)^{|S|}}{|S|+1} \prod_{s \in S} \Pr[X_s]$$

$$= p_r \sum_{\substack{S \subseteq \mathcal{R} \\ r \notin S}} \frac{(-1)^{|S|}}{|S|+1} \prod_{s \in S} p_s$$

$\square$

# B    Proofs for Section 3: Permute-and-Flip Mechanism

In this section, we first prove Proposition 2, which gives simplifed sufficient conditions for privacy for a regular mechanism. We then use Proposition 2 to prove Theorem 1, which establishes the privacy of permute-and-flip. Finally, we prove Proposition 3, which shows that permute-and-flip satisfies the recurrence used in the derivation.

**Proposition 2.** *A regular mechanism* $\mathcal{M} : \mathbb{R}^n \to \mathcal{R}$ *is* $\epsilon$-*differentially private if:*
$$\Pr[\mathcal{M}(\vec{q}) = r] \geq \exp(-\epsilon) \Pr[\mathcal{M}(\vec{q} + 2\Delta\vec{e}_r) = r]$$
*for all* $(\vec{q}, r)$, *where* $\vec{e}_r$ *is the unit vector with a one at position* $r$.

*Proof.* Let $\mathcal{M}$ be a regular mechanism satisfying:
$$\Pr[\mathcal{M}(\vec{q}) = r] \geq \exp(-\epsilon) \Pr[\mathcal{M}(\vec{q} + 2\Delta\vec{e}_r) = r] \tag{9}$$

Our goal is to show that $\mathcal{M}$ is differentially private, i.e., if for all $\vec{q} \in \mathbb{R}^n$, $r \in \mathcal{R}$, and $\vec{z} \in [-\Delta, \Delta]^n$,
$$\Pr[\mathcal{M}(\vec{q}) = r] \leq \exp(\epsilon) \Pr[\mathcal{M}(\vec{q} + \vec{z}) = r]$$

Using this assumption together with the regularity of $\mathcal{M}$, we obtain:

$$\begin{aligned}
\Pr[\mathcal{M}(\vec{q}) = r] &\leq \exp(\epsilon) \Pr[\mathcal{M}(\vec{q} - 2\Delta\vec{e}_r) = r] &&\text{by Equation (9)} \\
&= \exp(\epsilon) \Pr[\mathcal{M}(\vec{q} + \Delta\vec{1} - 2\Delta\vec{e}_r) = r] &&\text{by shift-invariance} \\
&\leq \exp(\epsilon) \Pr[\mathcal{M}(\vec{q} + \vec{z}) = r] &&\text{by monotonicity}
\end{aligned}$$

Thus, we conclude that $\mathcal{M}$ is differentially-private, as desired. This completes the proof.    $\square$

Before proving Theorem 1, we will argue regularity.

**Lemma 3.** $\mathcal{M}_{PF}$ *is regular.*

*Proof.* We will establish the three conditions: symmetry, shift-invariance, and monotonicity.

- **Symmetry**: Consider $p_r$ as defined in the definition of $\mathcal{M}_{PF}$, and let $\vec{p}' = \Pi\vec{p}$ denote the same vector on the permuted quality scores. Now note that every permutation is equally likely for both $\vec{p}$ and $\vec{p}'$, and that the only difference is that $p_r = p'_{\pi(r)}$. Hence $\Pr[\mathcal{M}(\vec{q}) = r] = \Pr[\mathcal{M}(\Pi\vec{q}) = \pi(r)]$, which implies $\mathcal{M}$ is symmetric as desired.

- **Shift-invariance**: $\mathcal{M}_{PF}$ is shift-invariant because on only depends on $\vec{q}$ through $q_r - q_*$. Adding a constant to $\vec{q}$ does not change $q_r - q_*$.

- **Monotonicity**: Monotonicity follows from the pmf of the mechanism:

$$\Pr[\mathcal{M}_{PF}(\vec{q}) = r] = p_r \sum_\pi \frac{1}{n!} \prod_{s:\pi(s)<\pi(s)} (1 - p_s)$$

.

Assume without loss of generality that $q_* = 0$ and note that $p_r = \exp\left(\frac{\epsilon}{2\Delta} q_r\right)$. Clearly, the expression above is monotonically increasing in $p_r$ (and hence $q_r$) and monotonically decreasing in $p_s$ (and hence $q_s$). Hence $\mathcal{M}_{PF}$ satisfies the monotonicity property.

Because $\mathcal{M}_{PF}$ is symmetric, shift-invariant, and monotonic, it is regular. □

**Theorem 1.** *The Permute-and-Flip mechanism $\mathcal{M}_{PF}$ is regular and $\epsilon$-differentially private.*

*Proof.* Lemma 3 established regularity. It remains to argue that $\mathcal{M}_{PF}$ is differentially-private. Let $\vec{q}$ and $r$ be arbitrary. By Proposition 2, it suffices to show that

$$\Pr[\mathcal{M}_{PF}(\vec{q}) = r] \geq \exp(-\epsilon) \Pr[\mathcal{M}_{PF}(\vec{q} + 2\Delta \vec{e}_r) = r]$$

or equivalently,

$$\log \Pr[\mathcal{M}_{PF}(\vec{q} + 2\Delta \vec{e}_r) = r] - \log \Pr[\mathcal{M}_{PF}(\vec{q}) = r] \leq \epsilon.$$

Assume without loss of generality that $\max_{s \neq r} q_s = 0$, so that $q_r$ is a maximum score if and only if $q_r \geq 0$. Let $f_r(\vec{q}) = \log \Pr[\mathcal{M}_{PF}(\vec{q}) = r]$. Then is enough to show that $\frac{\partial}{\partial q_r} f_r(\vec{q}) \leq \frac{\epsilon}{2\Delta}$ for all $\vec{q}$, since

$$\log \Pr[\mathcal{M}_{PF}(\vec{q} + 2\Delta \vec{e}_r) = r] - \log \Pr[\mathcal{M}_{PF}(\vec{q}) = r] = f_r(\vec{q} + 2\Delta \vec{e}_r) - f_r(\vec{q})$$
$$= \int_{q_r}^{q_r+2\Delta} \frac{\partial}{\partial q_r} f_r(\vec{q})\bigg|_{q_r=t} dt$$

The final equality is justified because, by the definition of the pmf for $\mathcal{M}_{PF}$, the function $f_r(\vec{q})$ is continuous. Furthermore, there is at most one point of non-differentiability of the partial derivative (at $t = 0$, when the $r$th score becomes equal to the maximum), so, if needed, the integral can be split into two parts about $t = 0$. This integral is bounded by $\epsilon$ as long the partial derivative $\frac{\partial f_r}{\partial q_r}$ is bounded by $\frac{\epsilon}{2\Delta}$.

Using the expression for the probability mass function of $\mathcal{M}_{PF}$ from Lemma 2, we have:

$$f_r(\vec{q}) = \log \Pr[\mathcal{M}_{PF}(\vec{q}) = r] = \log\left( p_r \sum_{\substack{S \subseteq \mathcal{R} \\ r \notin S}} \frac{(-1)^{|S|}}{|S|+1} \prod_{s \in S} p_s \right)$$

We will show using this formula that $\frac{\partial f_r}{\partial q_r}$ is always bounded by $\frac{\epsilon}{2\Delta}$. We examine the cases when $q_r < 0$ and $q_r \geq 0$ separately.

**Case 1:** $q_r < 0$. In this case, observe that $p_s = \exp\left(\frac{\epsilon}{2\Delta} q_s\right)$ does not depend on $q_r$ for $s \neq r$. Therefore, differentiating the formula for $f_r(\vec{q})$ gives

$$\frac{\partial f_r}{\partial q_r} = \frac{\partial f_r}{\partial p_r} \frac{\partial p_r}{\partial q_r}$$
$$= \left[ \frac{1}{\Pr[\mathcal{M}(\vec{q}) = r]} \sum_{\substack{S \subseteq \mathcal{R} \\ r \notin S}} \frac{(-1)^{|S|}}{|S|+1} \prod_{s \in S} p_s \right] \left[ p_r \frac{\epsilon}{2\Delta} \right]$$
$$= \frac{\epsilon}{2\Delta} \frac{\Pr[\mathcal{M}(\vec{q}) = r]}{\Pr[\mathcal{M}(\vec{q}) = r]} = \frac{\epsilon}{2\Delta}$$

**Case 2:** $q_r \geq 0$. In this case, because $q_r$ is the maximum score, we have $p_s = \exp\left(\frac{\epsilon}{2\Delta}(q_s - q_r)\right)$ for all $r$. We therefore proceed by differentiating $f_r(\vec{q})$ using this expression for $p_s$:

$$
\begin{aligned}
\frac{\partial f_r}{\partial q_r} &= \frac{1}{\Pr[\mathcal{M}_{PF}(\vec{q}) = r]} \frac{\partial}{\partial q_r} \Pr[\mathcal{M}_{PF}(\vec{q}) = r] \\
&= \frac{1}{\Pr[\mathcal{M}_{PF}(\vec{q}) = r]} \frac{\partial}{\partial q_r} \sum_{\substack{S \subseteq \mathcal{R} \\ r \notin S}} \frac{(-1)^{|S|}}{|S| + 1} \prod_{s \in S} \exp\left(\frac{\epsilon}{2\Delta}(q_s - q_r)\right) \\
&= \frac{1}{\Pr[\mathcal{M}_{PF}(\vec{q}) = r]} \frac{\partial}{\partial q_r} \sum_{\substack{S \subseteq \mathcal{R} \\ r \notin S}} \frac{(-1)^{|S|}}{|S| + 1} \exp\left(-|S|\frac{\epsilon}{2\Delta}q_r\right) \prod_{s \in S} \exp\left(\frac{\epsilon}{2\Delta}q_s\right) \\
&= \frac{1}{\Pr[\mathcal{M}_{PF}(\vec{q}) = r]} \sum_{\substack{S \subseteq \mathcal{R} \\ r \notin S}} \frac{(-1)^{|S|}}{|S| + 1} \exp\left(-|S|\frac{\epsilon}{2\Delta}q_r\right)\left[-|S|\frac{\epsilon}{2\Delta}\right] \prod_{s \in S} \exp\left(\frac{\epsilon}{2\Delta}q_s\right) \\
&= \frac{1}{\Pr[\mathcal{M}_{PF}(\vec{q}) = r]} \sum_{\substack{S \subseteq \mathcal{R} \\ r \notin S}} \frac{(-1)^{|S|}}{|S| + 1}\left[-|S|\frac{\epsilon}{2\Delta}\right] \prod_{s \in S} p_s \\
&= \left[\frac{\epsilon}{2\Delta}\right] \frac{-1}{\Pr[\mathcal{M}_{PF}(\vec{q}) = r]} \sum_{\substack{S \subseteq \mathcal{R} \\ r \notin S}} \frac{(-1)^{|S|}}{|S| + 1}|S| \prod_{s \in S} p_s
\end{aligned}
$$

We now seek to show that

$$
\frac{-1}{\Pr[\mathcal{M}_{PF}(\vec{q}) = r]} \sum_{\substack{S \subseteq \mathcal{R} \\ r \notin S}} \frac{(-1)^{|S|}}{|S| + 1}|S| \prod_{s \in S} p_s \leq 1
$$

Equivalently, by multiplying both sides by $\Pr[\mathcal{M}_{PF}(\vec{q}) = r]$ and rearranging, we would like to show:

$$
\Pr[\mathcal{M}_{PF}(\vec{q}) = r] + \sum_{\substack{S \subseteq \mathcal{R} \\ r \notin S}} \frac{(-1)^{|S|}}{|S| + 1}|S| \prod_{s \in S} p_s \geq 0
$$

Substituting the expression for $\Pr[\mathcal{M}_{PF}(\vec{q}) = r]$ and simplifying, the expression on the left-hand side above becomes:

$$
\begin{aligned}
\sum_{\substack{S \subseteq \mathcal{R} \\ r \notin S}} \frac{(-1)^{|S|}}{|S| + 1} \prod_{s \in S} p_s + \sum_{\substack{S \subseteq \mathcal{R} \\ r \notin S}} \frac{(-1)^{|S|}}{|S| + 1}|S| \prod_{s \in S} p_s &= \sum_{\substack{S \subseteq \mathcal{R} \\ r \notin S}} \frac{(-1)^{|S|}}{|S| + 1} \prod_{s \in S} p_s \left(1 + |S|\right) \\
&= \sum_{\substack{S \subseteq \mathcal{R} \\ r \notin S}} (-1)^{|S|} \prod_{s \in S} p_s \\
&= \prod_{s \in \mathcal{R} \setminus \{r\}} (1 - p_s)
\end{aligned}
$$

The final equality can be seen directly by multiplying out $\prod_{s \in \mathcal{R} \setminus \{r\}}(1 - p_s)$ or (equivalently) via the inclusion-exclusion formula. The final expression is the probability that the coins for all $s \in \mathcal{R} \setminus \{r\}$ are "tails", and is clearly non-negative, as desired.

This completes the proof.

$\square$

**Remark 1.** *When the quality function is* monotonic *in the sense that adding an individual to the dataset can only increase $q_r$ (and not decrease it), $\mathcal{M}_{PF}$ offers $\frac{\epsilon}{2}$-differential privacy. The proof is largely the same, but the worst-case neighbor from Proposition 2 now occurs when $\vec{q}' = \vec{q} + \Delta \vec{e}_r$.*

**Proposition 3.** $\mathcal{M}_{PF}$ *solves the recurrence relation in Equation* (6).

*Proof.* We proceed in cases:

**Case 1:** $q_r = q_*$

Because $\mathcal{M}_{PF}(\vec{q})$ is a valid probability distribution for all $\vec{q}$, and it is symmetric, it must satisfy case 2 of the recurrence relation.

**Case 2:** $q_r < q_*$

Note that the pmf of $\mathcal{M}_{PF}$ is:

$$\Pr[\mathcal{M}_{PF}(\vec{q}) = r] = p_r \sum_\pi \frac{1}{n!} \prod_{s:\pi(s)<\pi(r)} (1 - p_s)$$

$$\Pr[\mathcal{M}_{PF}(\vec{q} + (q_* - q_r)\vec{e}_r) = r] = p_r' \sum_\pi \frac{1}{n!} \prod_{s:\pi(s)<\pi(r)} (1 - p_s)$$

where $p_r = \exp\left(\frac{\epsilon}{2\Delta}(q_r - q_*)\right)$ and $p_r' = \exp\left(\frac{\epsilon}{2\Delta}(q_* - q_*)\right) = 1$. By comparing terms, it is clear that

$$\Pr[\mathcal{M}_{PF}(\vec{q}) = r] = \exp\left(\frac{\epsilon}{2\Delta}(q_r - q_*)\right) \Pr[\mathcal{M}_{PF}(\vec{q} + (q_* - q_r)\vec{e}_r) = r]$$

Hence, $\mathcal{M}_{PF}$ solves case 1 of the recurrence relation. This completes the proof.

$\square$

## C    Proofs for Section 4: Comparison with Exponential Mechanism

In this section, we first prove Theorem 2, which shows that the permute-and-flip error is no worse than the exponential mechanism for any score vector. We then prove Proposition 4 and Proposition 5, which analyze the worst-case expected errors of the two mechanisms and give tight lower bounds on expected error as the number of items $n$ increases.

### C.1    Proof of Theorem 2

We first prove two lemmas. The first lemma establishes a monotonicity property for the factor of the pmf from Lemma 1 *excluding $p_r$*, i.e., the function $g_r(\vec{q})$ such that $\Pr[\mathcal{M}_{PF}(\vec{q}) = r] = p_r \cdot g_r(\vec{q})$. The second lemma gives a useful fact about partial sums of a non-decreasing sequence.

**Lemma 4.** *If $q_r \leq q_s$, then $g_r(\vec{q}) \leq g_s(\vec{q})$, where*

$$g_r(\vec{q}) = \frac{1}{n!} \sum_\pi \prod_{t:\pi(t)<\pi(r)} (1 - p_t)$$

*Proof.* Recall that $p_r = \exp\left(\frac{\epsilon}{2\Delta}(q_r - q_*)\right)$. Note that if $q_r \leq q_s$ then $1 - p_r \geq 1 - p_s$. We will show that $g_s(\vec{q}) - g_r(\vec{q}) \geq 0$.

$$g_s(\vec{q}) - g_r(\vec{q}) = \frac{1}{n!} \sum_{\pi} \left[ \prod_{t:\pi(t)<\pi(s)} (1-p_t) - \prod_{t:\pi(t)<\pi(r)} (1-p_t) \right]$$

$$\overset{(a)}{=} \frac{1}{n!} \sum_{\pi:\pi(r)<\pi(s)} \left[ \prod_{t:\pi(t)<\pi(s)} (1-p_t) - \prod_{k:\pi(t)<\pi(r)} (1-p_t) \right]$$

$$+ \frac{1}{n!} \sum_{\pi:\pi(r)>\pi(s)} \left[ \prod_{t:\pi(t)<\pi(s)} (1-p_t) - \prod_{t:\pi(t)<\pi(r)} (1-p_t) \right]$$

$$\overset{(b)}{=} \frac{1}{n!} \sum_{\pi:\pi(r)<\pi(s)} \prod_{t:\pi(t)<\pi(s)} (1-p_t) - \frac{1}{n!} \sum_{\pi:\pi(r)>\pi(s)} \prod_{t:\pi(t)<\pi(r)} (1-p_t)$$

$$\overset{(c)}{=} \frac{1}{n!} \sum_{\pi:\pi(r)<\pi(s)} (1-p_r) \prod_{\substack{t:\pi(t)<\pi(s) \\ t\neq r}} (1-p_t) - \frac{1}{n!} \sum_{\pi:\pi(r)>\pi(s)} (1-p_s) \prod_{\substack{t:\pi(t)<\pi(r) \\ t\neq s}} (1-p_t)$$

$$\overset{(d)}{=} (p_s - p_r)\left[ \frac{1}{n!} \sum_{\pi:\pi(r)<\pi(s)} \prod_{\substack{t:\pi(t)<\pi(s) \\ t\neq s}} (1-p_t) \right]$$

$$\overset{(e)}{\geq} 0$$

Above, (a) breaks the sum up into permutations where $r$ precedes $s$ and vice versa. Step (b) cancels common terms (those that do not contain $1 - p_r$ or $1 - p_s$). Step (c) makes the dependence on $1 - p_r$ and $1 - p_s$ explicit. Step (d) rearranges terms and uses a variable replacement on the second sum (replacing $r$ with $s$). Step (e) uses the fact that both terms are non-negative. □

**Lemma 5.** *Let $\vec{f} \in \mathbb{R}^n$ be an arbitrary vector satisfying:*

1. *$f_1 \leq f_2 \leq \cdots \leq f_n$*

2. *$\sum_{r=1}^{n} f_r = 0$*

*Then for all $s = \{1, \ldots, n\}$, the following holds*

$$\sum_{r=1}^{s} f_r \leq 0$$

*Proof.* Let $m$ be any index satisfying $f_m \leq 0$ and $f_{m+1} \geq 0$. If $t \leq m$, the claim is clearly true, as it is a sum of non-positive terms. If $t > m$, we have $\sum_{r=1}^{s} f_r \leq \sum_{r=1}^{n} f_r = 0$. In either case the partial sum is non-positive, and the claimed bound holds. □

**Theorem 2.** $\mathcal{M}_{PF}$ *is never worse than* $\mathcal{M}_{EM}$. *That is, for all $\vec{q} \in \mathbb{R}^n$ and all $t \geq 0$,*

$$\mathbb{E}[\mathcal{E}(\mathcal{M}_{PF}, \vec{q})] \leq \mathbb{E}[\mathcal{E}(\mathcal{M}_{EM}, \vec{q})], \qquad \Pr[\mathcal{E}(\mathcal{M}_{PF}, \vec{q}) \geq t] \leq \Pr[\mathcal{E}(\mathcal{M}_{EM}, \vec{q}) \geq t]$$

*Proof.* We will prove the probability statement first, after which the expected error result will follow easily. Assume without loss of generality (by symmetry) that $q_1 \leq q_2 \leq \cdots \leq q_n$. Let $f_r(\vec{q}) = \Pr[\mathcal{M}_{PF}(\vec{q}) = r] - \Pr[\mathcal{M}_{EM}(\vec{q}) = r]$ and let $s$ denote the largest index satisfying $q_s \leq q_* - t$. Then $\Pr[\mathcal{E}(\mathcal{M}_{PF}, \vec{q}) \geq t] - \Pr[\mathcal{E}(\mathcal{M}_{EM}, \vec{q}) \geq t] = \sum_{r=1}^{s} f_r(\vec{q})$ and our goal is to show:

$$\sum_{r=1}^{s} f_r(\vec{q}) \leq 0$$

for all $s \in \{1, \ldots, n\}$. We first argue that $f_r$ monotonically increases with $q_r$, i.e., $f_1 \leq f_2 \leq \cdots \leq f_n$.

Note that $f_r(\vec{q})$ can be expressed as $p_r[g_r(\vec{q}) - h_r(\vec{q})]$, where

$$g_r(\vec{q}) = \frac{1}{n!} \sum_{\pi} \prod_{t:\pi(t)<\pi(r)} (1-p_t)$$

$$h_r(\vec{q}) = \frac{1}{\sum_{t\in\mathcal{R}} p_t}$$

Further, notice that the sequence $h_r$ (as $r$ ranges from 1 to $n$) is constant-valued, while, from Lemma 4, we know that $g_r$ is also non-decreasing. Thus the sequence $g_r - h_r$ is also non-decreasing. This, together with the fact that $p_r$ is non-negative and also non-decreasing, we know that $f_r$ is non-decreasing. This fact together with Lemma 5 shows $\sum_{r=1}^{s} f_r(\vec{q}) \le 0$, as desired.

The ordering of expected errors now follows directly. Specifically, the expected error can be expressed in terms of the (complementary) cumulative distribution function as:

$$\mathbb{E}[\mathcal{E}(\mathcal{M}, \vec{q})] = \int_0^\infty \Pr[\mathcal{E}(\mathcal{M}, \vec{q}) \ge t] dt.$$

We have already shown that $\Pr[\mathcal{E}(\mathcal{M}_{PF}, \vec{q}) \ge t] \le \Pr[\mathcal{E}(\mathcal{M}_{EM}, \vec{q}) \ge t]$. Thus:

$$\mathbb{E}[\mathcal{E}(\mathcal{M}_{PF}, \vec{q})] - \mathbb{E}[\mathcal{E}(\mathcal{M}_{EM}, \vec{q})] = \int_0^\infty \Pr[\mathcal{E}(\mathcal{M}_{PF}, \vec{q}) \ge t] - \Pr[\mathcal{E}(\mathcal{M}_{EM}, \vec{q}) \ge t] dt \le 0$$

Thus, we conclude $\mathbb{E}[\mathcal{E}(\mathcal{M}_{PF}, \vec{q})] \le \mathbb{E}[\mathcal{E}(\mathcal{M}_{EM}, \vec{q})]$, as desired.

$\square$

## C.2 Proofs for Worst-Case Error Analysis

**Proposition 4.** *The worst-case expected errors for both $\mathcal{M}_{EM}$ and $\mathcal{M}_{PF}$ occur when $\vec{q} = (c, \dots, c, 0) \in \mathbb{R}^n$ for some $c \le 0$. Let $p = \exp\left(\frac{\epsilon}{2\Delta} c\right)$. The expected errors for score vectors of this form are:*

$$\mathbb{E}[\mathcal{E}(\mathcal{M}_{EM}, \vec{q})] = \frac{2\Delta}{\epsilon} \log\left(\frac{1}{p}\right) \left[1 - \frac{1}{1 + (n-1)p}\right], \tag{7}$$

$$\mathbb{E}[\mathcal{E}(\mathcal{M}_{PF}, \vec{q})] = \frac{2\Delta}{\epsilon} \log\left(\frac{1}{p}\right) \left[1 - \frac{1 - (1-p)^n}{np}\right]. \tag{8}$$

*The worst-case expected errors are found by maximizing Equations* (7) *and* (8) *over $p \in (0, 1]$.*

*Proof.* Assume without loss of generality that $q_* = 0$ and note that $p_r = \exp\left(\frac{\epsilon}{2\Delta} q_r\right)$.

**Part 1:** $\mathcal{M}_{EM}$

The (negative) expected error of $\mathcal{M}_{EM}$ can be expressed as:

$$-\mathbb{E}[\mathcal{E}(\mathcal{M}_{EM}, \vec{q})] = -q_* + \sum_r q_r \frac{p_r}{\sum_s p_s}$$

$$= \frac{2\Delta}{\epsilon} \frac{1}{\sum_s p_s} \sum_r p_r \log(p_r)$$

Our goal is to show this is minimized when $p_1 = \cdots = p_{n-1}$. We procede by way of contradiction. Assume WLOG $p_1 < p_2$. We will argue that we can replace $p_1$ and $p_2$ with new values that decrease the objective. First write the negative expected error as a function of $p_1$ and $p_2$, treating everything else as a constant.

$$f(p_1, p_2) = \frac{1}{p_1 + p_2 + a} \left[p_1 \log(p_1) + p_2 \log(p_2) + b\right]$$

We will show that $f(\frac{p_1+p_2}{2}, \frac{p_1+p_2}{2}) < f(p_1, p_2)$.

$$f\left(\frac{p_1+p_2}{2}, \frac{p_1+p_2}{2}\right) = \frac{1}{2\frac{p_1+p_2}{2} + a}\left[2\frac{(p_1+p_2)}{2}\log\left(\frac{p_1+p_2}{2}\right) + b\right]$$

$$= \frac{1}{p_1+p_2+a}\left[2\frac{(p_1+p_2)}{2}\log\left(\frac{p_1+p_2}{2}\right) + b\right]$$

$$< \frac{1}{p_1+p_2+a}\left[p_1\log\left(p_1\right) + p_2\log\left(p_2\right) + b\right]$$

$$= f(p_1, p_2)$$

Above, the inequality follows from the strict convexity of $p\log(p)$. Thus, $f(p_1, p_2)$ is not a minimum, which is a contradiction.

Plugging in $p_n = 1$ and $p_r = p$ for $r < n$, we obtain:

$$\mathbb{E}[\mathcal{E}(\mathcal{M}_{EM}, \vec{q})] = -\frac{2\Delta}{\epsilon}\frac{(n-1)p\log p}{1+(n-1)p}$$

$$= -\frac{2\Delta}{\epsilon}\log(p)\frac{(n-1)p}{1+(n-1)p}$$

$$= -\frac{2\Delta}{\epsilon}\log(p)\left[1 - \frac{1}{1+(n-1)p}\right]$$

$$= \frac{2\Delta}{\epsilon}\log\left(\frac{1}{p}\right)\left[1 - \frac{1}{1+(n-1)p}\right]$$

**Part 2: $\mathcal{M}_{PF}$**

The (negative) expected error of $\mathcal{M}_{PF}$ can be expressed as:

$$-\mathbb{E}[\mathcal{E}(\mathcal{M}_{PF}, \vec{q})] = -q_* + \sum_r q_r p_r \sum_\pi \frac{1}{n!}\prod_{s:\pi(s)<\pi(r)}(1-p_s)$$

$$= \frac{2\Delta}{\epsilon}\sum_r p_r\log(p_r)\sum_\pi \frac{1}{n!}\prod_{s:\pi(s)<\pi(r)}(1-p_s)$$

We wish to show that this is minimized when $p_1 = \cdots = p_{n-1} = c$ for some $c \in (0, 1)$. We proceed by way of contradiction. Assume without loss of generality $p_1 < p_2$ and let $f(p_1, p_2) = -\mathbb{E}[\mathcal{E}(\mathcal{M}_{PF}, \vec{q})]$ be the negative expected error when treating everything constant except $p_1$ and $p_2$. Note that $f$ can be expressed as:

$$f(p_1, p_2) = p_1\log(p_1)[a(1-p_2)+b] + p_2\log(p_2)[a(1-p_1)+b]$$
$$- c(1-p_1) - c(1-p_2) - d(1-p_1)(1-p_2) - e$$

where $a, b, c, d, e \geq 0$. We proceed in cases, by showing that we can always find new values for $p_1$ and $p_2$ that reduces $f$

**Case 1:** $p_1\log(p_1) < p_2\log(p_2)$

Set $p_2 \leftarrow p_1$.

The second term in the sum is (strictly) less by the assumption of case 1. Every other term is strictly less because $p_1 < p_2$, which implies $(1-p_1) > (1-p_2)$ or equivalently $-(1-p_1) < -(1-p_2)$.

**Case 2:** $p_1\log(p_1) \geq p_2\log(p_2)$

Set $p_1 = p_2 \leftarrow \frac{p_1 + p_2}{2}$.

Consider breaking up the sum into two pieces; i.e., $f(p_1, p_2) = f_A(p_1, p_2) + f_B(p_1, p_2)$ where:

$$f_A(p_1, p_2) = p_1 \log (p_1)[a(1 - p_2) + b] + p_2 \log (p_2)[a(1 - p_1) + b]$$
$$f_B(p_1, p_2) = -c(1 - p_1) - c(1 - p_2) - d(1 - p_1)(1 - p_2) - e$$

We have:

$$
\begin{aligned}
f_A\left(\frac{p_1 + p_2}{2}, \frac{p_1 + p_2}{2}\right) &= 2\frac{p_1 + p_2}{2} \log \left(\frac{p_1 + p_2}{2}\right)\left[a\left(1 - \frac{p_1 + p_2}{2}\right) + b\right] \\
&= (p_1 + p_2) \log \left(\frac{p_1 + p_2}{2}\right)\left[a(1 - p_1) + b + a(1 - p_2) + b\right] \\
&< \frac{1}{2} \cdot \left[p_1 \log (p_1) + p_2 \log (p_2)\right]\left[a(1 - p_1) + b + a(1 - p_2) + b\right] \\
&= \frac{1}{2}\Big[p_1 \log (p_1)[a(1 - p_1) + b] + p_1 \log (p_1)[a(1 - p_2) + b] \\
&\quad + p_2 \log (p_2)[a(1 - p_1) + b] + p_2 \log (p_2)[a(1 - p_2) + b]\Big] \\
&< p_1 \log (p_1)[a(1 - p_2) + b] + p_2 \log (p_2)[a(1 - p_1) + b] \\
&= f_A(p_1, p_2)
\end{aligned}
$$

Above, the first step follows from linearity, and the second step follows from the convexity of $p \log (p)$ and non-negativeness of the linear term. The fourth step uses the assumption that $p_1 \log (p_1) \geq p_2 \log (\vec{p_2})$ (Case 2), and the fact that $a(1 - p_1) + b > a(1 - p_2) + b$ and $\log (p_2) < 0$.

$$
\begin{aligned}
f_B\left(\frac{p_1 + p_2}{2}\right) &= -2c\left(1 - \frac{p_1 + p_2}{2}\right) - d\left(1 - \frac{p_1 + p_2}{2}\right)^2 - e \\
&= -c(1 - p_1) - c(1 - p_2) - d\left(1 - \frac{p_1 + p_2}{2}\right)^2 - e \\
&< -c(1 - p_1) - c(1 - p_2) - d(1 - p_1)(1 - p_2) - e \\
&= f_B(p_1, p_2)
\end{aligned}
$$

Above, the first step follows from linearity, and the second step follows from the fact that the area of a square is always larger than the area of a rectangle with the same perimeter.

We have shown that $f_A$ and $f_B$ are both reduced, so $f$ as a whole is also reduced.

To derive the expected error for a quality score vector of this form, we use a simple probabilistic argument. There are $n - 1$ items with probability $p$ coins, and one item with a probability 1 coin. The probability of selecting an item corresponding to a probability $p$ coin is $\sum_{i=1}^{n} \frac{1}{n}(1 - (1 - p)^{i-1})$ where the index of the sum represents the location of the probability 1 item in the permutation and $1 - (1 - p)^{i-1}$ is the probability that at least one of the probability $p$ coins before position $i$ comes up heads. Using the formula for a geometic series, this simplifies to $1 - \frac{1 - (1 - p)^n}{np}$. Thus, recalling that $c = \frac{2\Delta}{\epsilon} \log (p)$, the expected error can be expressed as:

$$
\begin{aligned}
\mathbb{E}[\mathcal{E}(\mathcal{M}_{PF}, \vec{q})] &= c\left[1 - \frac{1 - (1 - p)^n}{np}\right] \\
&= -\frac{2\Delta}{\epsilon} \log (p)\left[1 - \frac{1 - (1 - p)^n}{np}\right] \\
&= \frac{2\Delta}{\epsilon} \log \left(\frac{1}{p}\right)\left[1 - \frac{1 - (1 - p)^n}{np}\right].
\end{aligned}
$$

This completes the proof. □

**Proposition 5.** *For $\vec{q} = (c, \ldots, c, 0) \in \mathbb{R}^n$ with $c = -\frac{2\Delta}{\epsilon} \log n$, the expected error $\mathbb{E}[\mathcal{E}(\mathcal{M}_{PF}, \vec{q})]$ of permute-and-flip is at least $\frac{\Delta}{2\epsilon} \log(n)$. This implies that $\mathbb{E}[\mathcal{E}(\mathcal{M}_{EM}, \vec{q})] \geq \frac{\Delta}{2\epsilon} \log(n)$ as well, and that the upper bounds of Proposition 1 and Corollary 1 are within a factor of four of being tight.*

Let $c = -\frac{2\Delta}{\epsilon} \log(n)$ and note that $p = \frac{1}{n}$ in Equation (8). Plugging in $p$ to Equation (8) and simplifying, we obtain:

$$
\begin{aligned}
\mathbb{E}[\mathcal{E}(\mathcal{M}_{PF}, \vec{q})] &= \frac{2\Delta}{\epsilon} \log(n) \left[ 1 - \frac{1 - (1 - \frac{1}{n})^n}{n^{\frac{1}{n}}} \right] \\
&= \frac{2\Delta}{\epsilon} \log(n) \left( 1 - \frac{1}{n} \right)^n \\
&\geq \frac{2\Delta}{\epsilon} \log(n) \frac{1}{4} \\
&= \frac{\Delta}{2\epsilon} \log(n)
\end{aligned}
$$

This completes the proof.

## D   Proofs for Section 5: Optimailty of Permute-and-Flip

In this section we prove Proposition 6, about Pareto optimality of permute-and-flip, and Theorem 3, about "overall" optimality.

**Proposition 6** (Pareto Optimality). *If $\mathbb{E}[\mathcal{E}(M_{PF}, \vec{q})] > \mathbb{E}[\mathcal{E}(\mathcal{M}, \vec{q})]$ for some regular mechanism $\mathcal{M}$ and some $\vec{q} \in \mathbb{R}^n_{2\Delta}$, then there exists $\vec{q}' \in \mathbb{R}^n_{2\Delta}$ such that $\mathbb{E}[\mathcal{E}(M_{PF}, \vec{q}')] < \mathbb{E}[\mathcal{E}(\mathcal{M}, \vec{q}')]$.*

*Proof.* Note that the expected error of the mechanism can be expressed as:

$$
\mathbb{E}[\mathcal{E}(\mathcal{M}, \vec{q})] = \sum_{\substack{r \in \mathcal{R} \\ q_r < q_*}} \Pr[\mathcal{M}(\vec{q}) = r](q_* - q_r)
$$

Since $\mathbb{E}[\mathcal{E}(\mathcal{M}_{PF}, \vec{q})] > \mathbb{E}[\mathcal{E}(\mathcal{M}, \vec{q})]$, then $\Pr[\mathcal{M}_{PF}(\vec{q}) = r] > \Pr[\mathcal{M}(\vec{q}) = r]$ for some $r$ where $q_r < q_*$. By Lemma 6, there must be some $\vec{q}'$ where $\mathbb{E}[\mathcal{E}(\mathcal{M}, \vec{q}')] > \mathbb{E}[\mathcal{E}(\mathcal{M}_{PF}, \vec{q}')]$. This completes the proof. □

**Lemma 6.** *If $\Pr[\mathcal{M}(\vec{q}) = r] < \Pr[\mathcal{M}_{PF}(\vec{q}) = r]$ for some $r$ where $q_r < q_*$, then there exists a $\vec{q}'$ such that $\mathbb{E}[\mathcal{E}(\mathcal{M}, \vec{q}')] > \mathbb{E}[\mathcal{E}(\mathcal{M}_{PF}, \vec{q}')]$.*

*Proof.* Let $\vec{q}' = \vec{q} + (q_* - q_r)\vec{e}_r$. By the differential privacy and regularity of $\mathcal{M}$ and the recursive construction of $\mathcal{M}_{PF}$, we know:

$$
\begin{aligned}
\Pr[\mathcal{M}(\vec{q}) = r] &\geq \exp\left( \frac{\epsilon}{2\Delta}(q_r - q_*) \right) \Pr[\mathcal{M}(\vec{q}') = r] \\
\Pr[\mathcal{M}_{PF}(\vec{q}) = r] &= \exp\left( \frac{\epsilon}{2\Delta}(q_r - q_*) \right) \Pr[\mathcal{M}_{PF}(\vec{q}') = r]
\end{aligned}
$$

Combining the above with the assumption of the Lemma, we obtain:

$$
\Pr[\mathcal{M}(\vec{q}) = r] < \Pr[\mathcal{M}_{PF}(\vec{q}) = r]
$$

Note that $\vec{q}'_r = \vec{q}'_*$. We proceed by way of induction:

**Base Case:** $n'_* = n - 1$.

There is a single $s$ such that $q'_s < q'_*$, and it must be the case that $\Pr[\mathcal{M}(\vec{q}') = s] > \Pr[\mathcal{M}_{PF}(\vec{q}') = s]$ by the symmetry and sum-to-one constraint on $\mathcal{M}$ and $\mathcal{M}_{PF}$. Thus, it follows immediately that

$\mathbb{E}[\mathcal{E}(\mathcal{M}_{PF}, \vec{q}')] < \mathbb{E}[\mathcal{E}(\mathcal{M}, \vec{q}')]$ because $\mathcal{M}$ places more probability mass on the candidate $s$ that increases the expected error (i.e., $q'_* - q'_s > 0$).

**Induction Step:** Assume Lemma 6 holds when $n'_* = k + 1$. We will show that Lemma 6 holds for $n'_* = k$.

We proceed in two cases:

**Case 1:** $\Pr[\mathcal{M}(\vec{q}') = s] \geq \Pr[\mathcal{M}_{PF}(\vec{q}') = s]$ for all $s$ such that $q'_s < q'_*$.

The inequality must be strict for some $s$, because the inequality is strict for all $r$ where $q_r = q_*$ by the regularity/symmetry of $\mathcal{M}$ and $\mathcal{M}_{PF}$. Thus, it follows immediately that $\mathbb{E}[\mathcal{E}(\mathcal{M}_{PF}, \vec{q}')] < \mathbb{E}[\mathcal{E}(\mathcal{M}, \vec{q}')]$ because $\mathcal{M}$ places more probability mass on the candidates $s$ that increase the expected error (i.e., $q'_* - q'_s > 0$).

**Case 2:** $\Pr[\mathcal{M}(\vec{q}') = s] < \Pr[\mathcal{M}_{PF}(\vec{q}') = s]$ for some $s$ such that $q'_s < q_*$.

Applying the induction hypothesis Lemma 6 using $\vec{q}'$ (now with $n'_* = k + 1$), we see that the claim must be true for $n'_* = k$, as desired.

$\square$

**Theorem 3** (Overall Optimality). *For all regular mechanisms $\mathcal{M}$ and all $\epsilon \geq \log\left(\frac{1}{2}(3 + \sqrt{5})\right)$,*

$$\sum_{\vec{q} \in Q} \mathbb{E}[\mathcal{E}(\mathcal{M}_{PF}, \vec{q})] \leq \sum_{\vec{q} \in Q} \mathbb{E}[\mathcal{E}(\mathcal{M}, \vec{q})]$$

*where $Q = \{\vec{q} \in \mathbb{R}^n_{2\Delta} : q_* - q_r \leq 2\Delta k, q_* = 0\}$ for any integer constant $k \geq 0$.*

For the above optimality criteria, the best mechanism can be obtained by solving a simple linear program. The variables of the linear program correspond to the probabilities the mechanism assigns to different $(\vec{q}, r)$ pairs, and the constraints are those required for differential privacy and regularity (which are all linear).

Denote the optimization variables as $x_r(\vec{q}) := \Pr[\mathcal{M}(\vec{q}) = r]$ for all $\vec{q} \in Q$ and all $r \in \mathcal{R}$. Then the linear program for the optimal regular mechanism can be expressed as:

$$
\begin{aligned}
\underset{x}{\text{maximize}} \quad & \sum_{\vec{q} \in Q} \sum_r x_r(\vec{q}) q_r \\
\text{subject to} \quad & x_r(\vec{q}) \geq \exp(-\epsilon) x_r(\vec{q}') && \forall \vec{q}, r && \text{(privacy)} \\
& x_r(\vec{q}) = x_{\pi(r)}(\Pi \vec{q}) && \forall \vec{q}, r, \pi && \text{(symmetry)} \\
& \sum_r x_r(\vec{q}) = 1 && \forall \vec{q} && \text{(sum-to-one)} \\
& x_r(\vec{q}) \geq 0 && \forall \vec{q}, r
\end{aligned}
$$

The first constraint enforces differential privacy for a regular mechanism as in Proposition 2, where $\vec{q}'$ is the worst-case neighbor of $\vec{q}$. We assumed the maximum entry of every score vector is zero, which is without loss of generality due to shift invariance. To ensure that $\vec{q}'$ has maximum entry zero, we use separate expressions for $\vec{q}'$ depending on whether or not $q_r = 0$:

$$\vec{q}' = \begin{cases} \vec{q} + 2\Delta \vec{e}_r & q_r < 0 \\ \vec{q} + 2\Delta(\vec{e}_r - \vec{1}) & q_r = 0 \end{cases}$$

The second constraint ensures the mechanism is symmetric, and the final two constraints ensure the mechanism corresponds to a valid probability distribution.

To measure how close to optimal permute-and-flip is for $\epsilon$ below the threshold, we can solve this linear program numerically, and compare the solution to permute-and-flip. Observe that the linear program has a large number of redundant variables from the symmetry constraint (e.g., $x_1(-2, -8, 0) = x_3(0, -8, -2)$). These variables can be grouped into equivalence classes, and the redundant ones can be eliminated, keeping only a single one from each equivalence class. This drastically reduces the number of variables and also allows us to eliminate the symmetry constraints. Using this trick, the resulting linear program is significantly smaller, but the size still grows quickly with $n$ and $k$, and is only feasible to solve for relatively small $n$ and $k$.

**Relaxed LP**   Our goal is to show that $\mathcal{M}_{PF}$ solves the linear program. To do so, we will consider the following relaxation of the linear program:

$$\underset{x}{\text{maximize}} \quad \sum_{\vec{q}\in Q}\sum_r x_r(\vec{q})q_r$$

$$\text{subject to} \quad -x_r(\vec{q}) + \exp\left(\frac{\epsilon}{2\Delta}q_r\right)x_r(\vec{q}-q_r\vec{e}_r) \le 0 \quad q_r < 0 \qquad\qquad \text{(privacy)}$$

$$n_* x_r(\vec{q}) + \sum_{s:q_s<0} x_s(\vec{q}) = 1 \qquad\qquad q_r = 0 \quad \text{(symmetry, sum-to-one)}$$

$$x_r(\vec{q}) \ge 0$$

In this linear program:

- There is exactly one constraint per optimization variable (excluding non-negativity constraints).
- The first set of constraints corresponds to a *subset* of the privacy constraints from the original, corresponding only to $(\vec{q}, r)$ pairs with $q_r < 0$. In addition, we performed substitutions of the form

$$\begin{aligned}
x_r(\vec{q}) &\ge \exp\left(-\epsilon\right)x_r(\vec{q}+2\Delta\vec{e}_r) \\
&\ge \exp\left(-2\epsilon\right)x_r(\vec{q}+4\Delta\vec{e}_r) \\
&\ge \cdots \\
&\ge \exp\left(\frac{\epsilon}{2\Delta}q_r\right)x_r(\vec{q}-q_r\vec{e}_r),
\end{aligned}$$

where $\vec{q}-q_r\vec{e}_r$ is the quality score vector obtained by setting $q_r = 0$.

- The sum-to-one and symmetry constraints are merged into a single constraint when $q_r = 0$, and other symmetry constraints are dropped.

These constraints correspond exactly to the ones in the recurrence defining $\mathcal{M}_{PF}$ in Section 3.1. This means that $\mathcal{M}_{PF}$ satisfies these constraints with equality, by construction. Furthermore, since $\mathcal{M}_{PF}$ is feasible in the full LP (because it is a private, regular mechanism), if $\mathcal{M}_{PF}$ is optimal for the relaxed LP it is also optimal for the full LP.

**Constructing a dual optimal solution**   We can show that $\mathcal{M}_{PF}$ is optimal by constructing a corresponding optimal solution to the dual linear program:

$$\underset{y}{\text{minimize}} \quad \sum_{\vec{q}}\sum_{r:q_r=0} y_r(\vec{q})$$

$$\text{subject to} \quad n_* y_r(\vec{q}) - \sum_{t=1}^{k} y_r(\vec{q}-2\Delta t\vec{e}_r)\exp\left(-t\epsilon\right) \ge q_r \quad q_r = 0$$

$$-y_r(\vec{q}) + \sum_{s:q_s=0} y_s(\vec{q}) \ge q_r \qquad\qquad q_r < 0$$

$$y_r(\vec{q}) \ge 0 \qquad\qquad q_r < 0$$

Because there is exactly one constraint for each optimization variable, we have used the same indexing scheme for the dual variables. Note that the non-negativity constraints apply only to $(\vec{q}, r)$ pairs with $q_r < 0$.

To prove optimality, the dual solution and $\mathcal{M}_{PF}$ should satisfy complementary slackness: for each positive primal variable, the corresponding dual constraint should be tight. However, *all* primal variables are positive. Therefore, all dual constraints must be tight. By treating dual constraints as equalities, we obtain a recurrence for $y$ similar to the one used to derive $\mathcal{M}_{PF}$:

$$y_r(q) = \begin{cases} 0 & q_r = 0, n_* = 1 \\ -\frac{1}{n_*}\sum_{t=1}^{k} y_r(\vec{q}-2\Delta t\vec{e}_r)\exp\left(-t\epsilon\right) & q_r = 0 \\ -q_r + \sum_{s:q_s=0} y_s(q) & q_r < 0 \end{cases}$$

Like the recurrence for $\mathcal{M}_{PF}$, this recurrence is well-founded and defines a unique dual solution $y$. The order of evaluation is reversed for the dual variables, and the base case occurs when $n_* = 1$

(rather than $n_* = n$). We will now argue that, whenever $\epsilon \geq \log\left(\frac{1}{2}(3 + \sqrt{5})\right)$, the resulting dual solution is feasible. This, together with complementary slackness, which is satisfied by construction, implies that $\mathcal{M}_{PF}$ and $y$ are optimal solutions to the primal and dual programs, respectively.

Let $y$ solve the recurrence above for $\epsilon \geq \log\left(\frac{1}{2}(3 + \sqrt{5})\right)$. To show that $y$ is feasible, we will argue inductively that these finer-grained bounds hold:

$$-\frac{2\Delta}{n_*} \leq y_r(\vec{q}) \leq 0 \qquad\qquad \text{if } q_r = 0 \qquad\qquad (10)$$

$$0 \leq y_r(\vec{q}) \leq -q_r \qquad\qquad \text{if } q_r < 0 \qquad\qquad (11)$$

Note that Equation (11) includes the dual feasibility constraints.

We prove Equations (10) and (11) by induction on the $n_*$, the number of zero (i.e., maximum) entries of $\vec{q}$. For the base case, when $n_* = 1$, $y_r(q) = -q_r$, so Equations (10) and (11) hold.

Now let $\vec{q}$ be a score vector with $n_* > 1$ entries equal to zero, and assume that Equations (10) and (11) hold for all score vectors with fewer than $n_*$ zeros. By Case 1 of the recurrence, for $r$ such that $q_r = 0$, we have

$$y_r(\vec{q}) = -\frac{1}{n_*} \sum_{t=1}^{k} y_r(\vec{q} - 2\Delta t\vec{e}_r) \exp(-t\epsilon)$$

$$\geq -\frac{1}{n_*} \sum_{t=1}^{k} 2\Delta t \exp(-t\epsilon)$$

$$\geq -\frac{2\Delta}{n_*} \sum_{t=1}^{\infty} t \exp(-t\epsilon)$$

$$\geq -\frac{2\Delta}{n_*}.$$

In the second line, we used the fact that $y_r(\vec{q} - 2\Delta t\vec{e}_r) \leq -(\vec{q} - 2\Delta t\vec{e}_r)_r = 2\Delta t$, which follows from Equation (11) by the induction hypothesis, since $\vec{q} - 2\Delta t\vec{e}_r$ is a score vector with $n_* - 1$ zeros. In the third line, we used the fact that $\sum_{t=1}^{\infty} t\exp(-t\epsilon) \leq 1$ whenever $\epsilon \geq \log\left(\frac{1}{2}(3 + \sqrt{5})\right)$, which is stated and proved in Lemma 7 below.

It is also clear that

$$y_r(\vec{q}) = -\frac{1}{n_*} \sum_{t=1}^{k} y_r(\vec{q} - 2\Delta t\vec{e}_r) \exp(-t\epsilon) \leq 0,$$

since, again by Equation (11) and the induction hypothesis, each term of the sum is non-nonegative.

We have now established that Equation (10) holds for all score vectors with $n_*$ or fewer zeros, which we use to prove that Equation (11) holds under the same conditions. By Case 2 of the recurrence, when $q_r < 0$ we have

$$y_r(\vec{q}) = -q_r + \sum_{s:q_s=0} y_s(\vec{q})$$

$$\geq -q_r + \sum_{s:q_s=0} -\frac{2\Delta}{n_*}$$

$$\geq -q_r - 2\Delta$$

$$\geq 0.$$

In the second line, we used, from Equation (10) that $y_s(\vec{q}) \geq -\frac{2\Delta}{n_*}$. Similarly, we have

$$y_r(\vec{q}) = -q_r + \sum_{s:q_s=0} y_s(\vec{q}) \leq -q_r$$

because $y_s(\vec{q}) \leq 0$.

This completes the inductive proof, and establishes that the dual solution $y$ is feasible. This in turn completes the proof that $\mathcal{M}_{PF}$ is optimal.

**Lemma 7.** *If $\epsilon \geq \log\left(\frac{1}{2}(3 + \sqrt{5})\right)$, then $\sum_{k=1}^{\infty} k \exp(-k\epsilon) \leq 1$.*

*Proof.* The infinite sum is equal to:

$$\frac{\exp(\epsilon)}{[1 - \exp(\epsilon)]^2}$$

Making the substitution $\exp(\epsilon) = 1 + z$, we have:

$$\frac{\exp(\epsilon)}{[1 - \exp(\epsilon)]^2} \leq 1 \iff \frac{1+z}{z^2} \leq 1 \iff 1 + z \leq z^2$$

The solution to the quadratic equation $1 + z = z^2$ is the golden ratio, $\phi = \frac{1}{2}(1 + \sqrt{5})$, so the inequality holds whenever $z \geq \phi$, or whenever $\epsilon \geq \log(1 + \phi) = \log\left(\frac{1}{2}(3 + \sqrt{5})\right)$. $\qquad\square$

## E   Dynamic Programming Algorithm

In this section, we derive an efficient $O(n^2)$ dynamic programming algorithm to calculate the probabilities. Recall the expression for the pmf from Lemma 2:

$$Pr[\mathcal{M}_{PF}(\vec{q}) = r] = p_r \sum_{\substack{S \subseteq \mathcal{R} \\ r \notin S}} \frac{(-1)^{|S|}}{|S| + 1} \prod_{s \in S} p_s.$$

To evaluate the probabilities efficiently, we can break up the sum into groups where $|S| = k$. Then, using dynamic programming, we can calculate these sums efficiently and use them to compute the desired probabilities.

Let

$$S(k, r) = \sum_{\substack{S \subseteq \mathcal{R} \\ |S| = k \\ \max(S) \leq r}} \prod_{s \in S} p_s.$$

And note that $S(k, r)$ satisfies the recurrence:

$$S(k, r) = S(k, r - 1) + p_r S(k - 1, r - 1).$$

$S(k, r - 1)$ is the sum over subsets not including $r$, and $p_r S(k - 1, r - 1)$ is the sum over subsets including $r$. Using the above recursive formula together with the base cases $S(0, r) = 1$ and $S(k, 0) = 0$, we can compute $S(k, r)$ for all $(k, r)$ in $O(n^2)$ time.

$S(k, n)$ is then the sum over all subsets of size $k$. Let $T(k, r)$ denote the sum over all size $k$ subsets *not including $r$*:

$$T(k, r) = \sum_{\substack{S \subseteq \mathcal{R} \\ |S| = k \\ r \notin S}} \prod_{s \in S} p_s.$$

and note that $T(k, r)$ satisfies the recurrence:

$$T(k, r) = S(k, n) - p_r T(k - 1, r)$$

with $T(0, r) = 1$. $T(k, r)$ can also be calculated in $O(n^2)$ time. The final answer is then:

$$\Pr[\mathcal{M}_{PF}(\vec{q}) = r] = p_r \sum_{k=0}^{n} \frac{(-1)^k}{k+1} T(k, r)$$

which can be computed in $O(n)$ time for each $r$. Thus, the overall time complexity of this dynamic programming procedure is $O(n^2)$.

## F   Report Noisy Max

A popular alternative to the exponential mechanism for private selection is report noisy max, which works by adding Laplace noise with scale $\frac{2\Delta}{\epsilon}$ to the score for each candidate, then returns the candidate with the largest noisy score.

Reasoning about report noisy max analytically and exactly is challenging, and we are not aware of a simple closed form expression for its probability mass function. To compute the probability of returning a particular candidate, we must reason about the probability that one random variable (the noisy score for that candidate) is larger than $n - 1$ other random variables (the scores for other candidates), which in general requires evaluating a complicated integral. Specifically, let $f(x)$ denote the probability density function of $\mathrm{Lap}(\frac{2\Delta}{\epsilon})$ and let $F(x)$ denote its cumulative density function.

$$\Pr[\mathcal{M}_{NM}(\vec{q}) = r] = \int_{-\infty}^{\infty} f(x) \prod_{s \neq r} F(q_r - q_s + x) dx$$

If we consider quality score vectors of the form $\vec{q} = (c, \ldots, c, 0)$, the expression simplies to:

$$\Pr[\mathcal{M}_{NM}(\vec{q}) = n] = \int_{-\infty}^{\infty} f(x) F(x - c)^{n-1} dx$$

Due to symmetry, the expected error can be expressed as:

$$\mathbb{E}[\mathcal{E}(\mathcal{M}_{NM}, \vec{q})] = -c\Big(1 - \Pr[\mathcal{M}_{NM}(\vec{q}) = n]\Big)$$

While it is not obvious how to simplify this expression further, we can readily evaluate the integral numerically to obtain the expected error. Doing so allows us to compare report noisy max with the exponential mechanism and permute-and-flip. Figure 4 plots the expected error of report noisy max alongside the exponential mechanism and permute-and-flip for quality score vectors of the form $\vec{q} = (c, c, 0)$. It shows that report noisy max is better than the exponential mechanism when $c$ is closer to $0$ but is worse when $c$ is much smaller than $0$. We made similar observations for different values of $n$ as well. Thus, we conclude that neither one Pareto dominates the other. On the other hand, permute-and-flip is always better than both mechanisms for all $c$. Note that in contrast to Figure 1a, we plot $c$ on the x-axis instead of $p = \exp\left(\frac{\epsilon}{2\Delta}c\right)$, because it is not clear if report noisy max only depends on $c$ through $p$.

This comparison covers a particular class of quality score vectors which allow for a simple and tractable exact comparison. Further comparison with report noisy max would be an interesting future direction.

## G   Extra Experiments

In Figure 5 and Figure 6, we measure the expected error of $\mathcal{M}_{EM}$ and $\mathcal{M}_{PF}$ on the mode and median problem for five different datasets from the DPBench study [20]. The conclusions are the same for each dataset: the improvement increases with $\epsilon$, and for the range of $\epsilon$ that offer reasoanble utility, the improvement is close to $2\times$. In Figure 7, we compare the expected error of $\mathcal{M}_{EM}$ and $\mathcal{M}_{PF}$ on both problems, for the value of $\epsilon$ satisfying $\mathbb{E}[\mathcal{E}(\mathcal{M}_{EM}, \vec{q})] = 50$.

Figure 4: Expected error of three mechanisms on quality score vectors of the form $\vec{q} = (c, c, 0)$ assuming $\epsilon = 1.0$ and $\Delta = 1.0$.

(a) HEPTH     (b) ADULTFRANK     (c) MEDCOST     (d) SEARCHLOGS     (e) PATENT

Figure 5: Expected error of $\mathcal{M}_{EM}$ and $\mathcal{M}_{PF}$ on five datasets for the mode problem.

(a) HEPTH     (b) ADULTFRANK     (c) MEDCOST     (d) SEARCHLOGS     (e) PATENT

Figure 6: Expected error of $\mathcal{M}_{EM}$ and $\mathcal{M}_{PF}$ on five datasets for the median problem.

(a) Mode                                (b) Median

Figure 7: Expected error of $\mathcal{M}_{EM}$ and $\mathcal{M}_{PF}$ on five datasets for both problems.