[Reviews · NeurIPS 2020]

Review 1

Summary and Contributions: This paper studies the selection problem, which (in the most general case) can be stated as follows: there is a data-independent set of n candidate solutions R that we would like to select from. For each candidate r in R, there is a quality score function q_r that maps any input dataset D to a real number q_r(D). The goal is to, given dataset D, select r in R that maximizes q_r(D), while respecting the notion of differential privacy (DP). Here, we say that the algorithm incurs an error of q^* - Expectation[q_{output}(D)] where q^* = min_r q_r(D); in other words, the error is the expected quality loss of the return solution compared to the optimum. Many well-studied problems in machine learning can be stated in the selection formulation; for example, each r could be a hypothesis and q_r(D) is the empirical error. A standard solution to this problem is the so-called Exponential Mechanism (McSherry and Talwar, FOCS'07) which works as follows. Let Delta be the sensitivity of q_r, i.e. the maximum change of q_r when changing a single user's input in the dataset D. To get an epsilon-DP algorithm, we output each r in R with probability proportional to exp(epsilon * q_r(D) / (2 * Delta)). The main contributions to this paper are as follows. (i) The authors propose a mechanism, called Permute-and-Flip (PF) that provably improves on Exponential Mechanism (EM) in the following sense. First, the expected error PF is never larger than that of EM with respect to any dataset and any scoring function. Second, for the dataset that gives worst case error, PF provably gives a smaller error than EM; the improvement depends on n, epsilon, Delta but it is noticeable as long as n, epsilon, Delta are moderate values. Moreover, in extreme datasets where one solution is really good (high quality score) and all other solutions are really bad (very low quality score), the improvement can be as larger as twice. To describe a PF, it is useful to first restate EM in the following form. Repeat the following until we output a solution: randomly pick a solution r from R, then with probability exp(epsilon / 2Delta * (q_r(D) - q^*)) output r. With this perspective, PF can be viewed as a without-replacement variant of EM. Specifically, instead of randomly pick r from R in each step, we randomly permute R beforehand. We then enumerate through each candidate r using this ordering, and we output r with probability exp(epsilon / 2Delta * (q_r(D) - q^*)). (ii) The authors show that the PF is provably optimal for epsilon >= 2.62 among all regular mechanisms that satisfy "reasonable" properties of symmetry, shift invariants and monotonicity. The authors show this by writing out the linear program (LP) that describes epsilon-DP constraints and give a dual solution that exactly matches the error of PF. In the case of small epsilon, the authors numerically solves the LP and shows that, in the case where epsilon << 1, there seems to be better protocol than both PF. Nonetheless, the gap in the performance is quite small (~ 10%) for moderate value of n and the interested "region" (i.e. k in the paper). (iii) Finally, the authors perform experiments on real-world datasets. Specifically, they show that, when on computing the mode and median of these datasets, PF gives almost 2x decrease in the error compared to EM. This demonstrates the potential effectiveness of PF in practice as well.

Strengths: - The exponential mechanism, together with Additive noise mechanisms (e.g. Laplace, Gaussian), is among the most widely used building block in differential privacy, both in theory and in practice. As a result, an improvement to the exponential mechanism could have a wide reaching impact (both in theory and practice). - The proposed Permute-and-Flip mechanism and its analysis are both simple and elegant. It could also be possible that the techniques are useful elsewhere. - The paper is well written and easy to follow/understand.

Weaknesses: I do not see a significant weakness of the paper.

Correctness: Since most of the proofs are in the appendix, I did not verify them in detail. However, the main ideas & proof overviews presented in the main body make sense and look correct.

Clarity: Yes, the theorem statements are clear and easy to follow. One possible improvement here is to discuss more about the "optimal" mechanism found by linear program (line 220-225). For example, if I understand correctly, the linear program will give probability masses only on the 2Delta-lattice; is it true that one can extend it to get an epsilon-DP algorithm for the entire R^n region? I did not find any of this detail even in the supplementary material. Update after author's response: The authors point out that the question of extending LP to the entire R^n region is an open question and will highlight this in the revision. This is a satisfactory explanation and hence I keep the same score as before.

Relation to Prior Work: Yes, the paper clearly discussed the Exponential Mechanism and compare it to the proposed Permute-and-Flipped Mechanism.

Reproducibility: Yes

Additional Feedback: Update after author's response:


Review 2

Summary and Contributions: This paper presents the novel "permute and flip" algorithm for differentially private selection. The main prior approach was the exponential mechanism which assigns scores to all possible outcomes, and performs biased sampling, exponentially weighted by score. The permute and flip algorithm instead randomly permutes the set of possible outputs, and performs bernoulli sampling with success probability exponentially dependent on the score.

Strengths: +novel algorithm for private selection +faster expected run time than prior mechanisms for this, and better performance (according to a specific accuracy metric)

Weaknesses: -the accuracy metric is not the standard one -no analysis of run time improvements

Correctness: Yes

Clarity: Yes

Relation to Prior Work: Yes

Reproducibility: Yes

Additional Feedback: I think this new algorithm is fantastic, but not for the reasons emphasized in the paper. The authors highlight that the benefit of the P&F algorithm is that it has better performance according the metric of expected additive error w.r.t. the optimum. However, a more standard metric for EM is high probability additive error w.r.t. the optimum, e.g., as in MT07 and DR14. Is there a practical reason for considering this alternative metric? No high-probability performance results were given for P&F, which left me wondering how the algorithm performs according to this metric. Do the authors have any results here? Relatedly, there is a subtle but important distinction that should be made clear in the paper. Technically, the EM is a canonical (eps,0)-DP mechanism, so any other (eps,0)-DP mech can be written as an instantiation of the EM with an appropriate score function. So when comparing P&F to EM and talking about Pareto improvements over EM, it's important to clarify that we're not talking about the EM in general, but rather EM with the scoring rule corresponding to the selection problem. This is related because I wonder what the corresponding EM score function be for P&F? What would the high probability guarantees of that look like? Perhaps one interpretation of the results in this paper is that when running the EM for private selection, an analyst can get better performance (according to which metric??) by using a non-standard scoring rule, specifically the one corresponding to the P&F algorithm. In my opinion, the main advantage of permute-and-flip is it's average case runtime, relative to the Exponential Mechanism. EM takes time linear in the size of the selection domain (both in the worst case and the average case), since it must compute the score for every possible outcome before it can perform the sampling. Although linear sounds good, the selection domain is usually extremely large in most interesting cases (e.g., SmallDB where the selection domain is exponentially sized w.r.t. the problem parameter), so the linear time of EM becomes infeasible in practice for these applications. Although P&F also takes linear time in the worst case, it will take much less time on average, or for any typical run. I would be extremely interested to see results along these lines for the P&F mechanism. E.g., does it make SmallDB feasible for practical use? How long does P&F take for some "typical" class of selection problems with very large domains? This could be tremendously valuable in terms of making private selection over large domains practical. The experimental results in the paper consider very small domain size--e.g., Figure 2 varies domain size between 2 and 6. Other comments (both strengths and weaknesses): -lines 45-46: the requirement that r approximately maximizes q_r(D) should be stated formally. In general, the notation that doesn't take r as an explicit input and suppresses dependence on D is non-standard, but I see why it greatly simplifies presentation here. It just requires a little extra care to make sure things are precise and clear. Similarly, in Def 3 I felt that there was missing a "\forall D" quantifier, but I guess it wasn't needed since we are implicitly encoding D in q. -I liked the axiomatic characterization of "reasonable" selection mechanisms. I think this may be of independent interest for future work on mechanisms for private selection. -I liked the comparison of Algorithms 2 and 3 side-by-side. This presentation and conceptualization was very clear in highlighting the differences between EM and P&F. -Thm 6 should specify q \in R^n. Otherwise it's not clear where the n comes from in the bounds. -Thm 8 and Figure 2: What is the role of k? It's included in the theorem statement but doesn't appear in the bounds, and then Figures 2bc show performance for varying values of k. But k is never explained, discussed, or intuited for the reader. -Broader impact is only one sentence, and the authors could provide a more thoughtful discussion here. ***EDIT: Thanks to the authors for addressing my concerns and comments, in particular the high-probability guarantees, and the other more subtle points. I have raised my score accordingly.


Review 3

Summary and Contributions: Permute-and-Flip is a novel mechanism for differentially private selection of elements based on a quality score and thus a direct competitor of the exponential mechanism and noisy-max. The mechanism works by simply permuting all elements and then iterating through the list, flipping biased coins for every element until the flip is successful (at which point this element is released). The probabilities are defined similarly to the ones used by the exponential mechanism, but shifted s.t. the element with the highest score has probability 1. Thus, the mechanism surely terminates and it can achieve better utility than the exponential mechanism in many cases.

Strengths: What I really like about the paper is the sheer simplicity of the mechanism. This approach can be easily implemented and used, which makes it a very valuable contribution to the field. I can see myself using Permute-and-Flip for my own future research. Making a good (in terms of utility) private selection of elements is an important problem that affects a lot of tasks, machine learning and otherwise. I think the paper is very relevant. The paper provides evidence for its improvement over the exponential mechanism in relevant privacy regimes.

Weaknesses: The paper only compares itself with the exponential mechanism. While that one is certainly popular, it's by no means the only mechanism for private selection (as the authors acknowledge in their discussion). I think that Permute-and-Flip should be compared with more mechanisms, including, prominently, the noisy max selection. As the exponential mechanism is at least one state-of-the-art mechanism and this one is shown to be strictly better, I don't necessarily see this as a reason for rejection.

Correctness: The claims are shown rigorously and appear to be sound.

Clarity: The paper is somewhat difficult to read due to the strong theoretical emphasis. I think adding a few more intuitions to the texts could help.

Relation to Prior Work: The authors do a good job in comparing their method with the exponential mechanism. It would be neat if they also compared against the Report Noisy Max mechanism for a couple of choices of noise (specifically: Laplace and Gauss).

Reproducibility: Yes

Additional Feedback: I don't think the emphasis on optimality for epsilon > 0.96 is particularly relevant as this is at the upper border of what I'd consider reasonable privacy. What I like, though, is that even for high-privacy regimes your approach significantly outperforms the exponential mechanism. Update after rebuttal: I liked the paper before and I still like it. With the added details mentioned in the rebuttal and some rephrasing of the text to make it more intuitive, this should be a strong paper.


Review 4

Summary and Contributions: This submission presents an enhanced version of exponential mechanism, to solve the problem of differentially private item selection. The key idea is to adopt sample-without-replacement instead of sample-with-replacement as is done in exponential replacement. The authors then prove two important properties: 1) Their new approach is always no worse than original exponential mechanism; 2) their approach is pareto-optimal in the sense that no other mechanism could outperform it in all settings.

Strengths: 1. This paper targets at an important problem in machine learning and data mining under differential privacy. 2. The method is very simple yet effective. 3. Solid analysis on the effectiveness of the proposed approach.

Weaknesses: N.A.

Correctness: Yes.

Clarity: This paper is well written and very easy to understand.

Relation to Prior Work: Yes. The connection to existing works is explained very clearly.

Reproducibility: Yes

Additional Feedback: I personally like this work very much. Exponential mechanism has been used extensively since the invention of DP, which converts numerical evaluation to categorical decision making space. It's a very important component to almost every system using DP, such as loss evaluation in machine learning and frequent pattern search in data mining. This work shows that it can be further improved. While the authors' experiments mainly focus on evaluating the effectiveness on simulated data, I believe it will be very interesting if the authors could take steps forward and see if it can help reduce the error of DP-based machine learning and data mining.

[Author Response · NeurIPS 2020]

We thank the reviewers for their thoughtful comments and diverse perspectives. The suggestions are excellent and will lead to numerous improvements to the paper. We will answer some of the questions directly in the paper, and state others as interesting open problems for future work. See below for our detailed plan.

**[R1, Optimal Mechanisms]** The reviewer makes a good observation by pointing out that the solution to the linear program does not define full mechanism, but just the probabilities of a mechanism on the $2\Delta$-lattice. It may have been a slight abuse of terminology to refer to the solution of the linear program as a "mechanism", so we will add in some supporting text to clarify this point.

It is not clear how one could extend the linear programming approach to define a full mechanism, as that would result in an infinite number of variables and constraints. It would be very nice if we could characterize the optimal mechanism using expected error *integrated* over all quality score vectors within some fixed box (the continuous analog of our current result), or even precisely quantify/bound the optimality gap for permute-and-flip under this objective. We don't know the answer to these questions, but will highlight it as an open problem.

**[R2, High probability error]** We thank the reviewer for encouraging us to think about this question. We acknowledge that the expected additive error is not the only relevant metric, but do point out that the main theorems of MT07 (e.g., Thm 8) do deal with expected error as we've defined it, although agree that DR14 derives a high-probability additive error bound instead.

We feel that expected error is a natural metric to optimize, although recognize that it is still useful to have guarantees w.r.t. the high probability error metric as well. We are unsure in general which of our results (e.g. optimality) may have analogs for high-probability additive error. However, we can easily show that PF inherits the guarantees of the exponential mechanism w.r.t any high probability error metric, much in the same way that it inherits the bound on expected error.

A little more precisely, we can show that PF *stochastically dominates* EM, that is:

$$\Pr[q_{PF} \geq x] \geq \Pr[q_{EM} \geq x]$$

for all $x$, where $q_{PF}$ and $q_{EM}$ are shorthand notation for $q_r$ where $r \sim M_{PF}(q)$ and $r \sim M_{EM}(q)$ respectively.

This follows directly from the Proof of Lemma 1, where we argue that $\Pr[\mathcal{M}_{PF} = r] - \Pr[\mathcal{M}_{EM} = r]$ monotonically increases with $q_r$.

Raising this question allowed us to derive a stronger statement about the performance of PF and simplify the existing proof of Theorem 5 at the same time, so thank you R2!

**[R2, Universality of EM]**

We will clarify the distinction of EM as a universal mechanism vs. EM for selection, as we use here. The reviewer raises an interesting question about non-standard scoring rules, which can hopefully be further explored and answered in future work.

**[R2, Expected Runtime]**

The reviewer rasises an interesting point on something we did not emphasize in this paper. We had previously thought a little about this idea, and concluded that for it to work, we need to know $q_*$ a-priori or have a sublinear time algorithm to find it. In these special cases, it would be possible to realize computational savings. We will state this as an open problem in the paper.

**[R3, Report Noisy Max]**

This is a good point. Report noisy max is tricky to reason about exactly because evaluating the probability mass function requires computing an $(n-1)$-dimensional integral in general, as it is the probability that one noisy answer is larger than $n-1$ others.

In the preliminary experiments hinted at in the related work, we studied the expected error of report noisy max in the special case where quality score vectors take on the form $(c, ..., c, 0)$. We will add those details and this experiment into the supplementary material and the full version of this paper, but leave further comparisons to future work.

**[R4, Error reduction of other mechanisms]**

This is another excellent idea, and something that we feel would greatly improve experiments. It would require additional space, so we state it as an important open problem.

[Meta-Review · NeurIPS 2020]

All four reviewers support acceptance of the paper. They argue that the permute and flip mechanism is simple and improves over the exponential mechanism for private selection in terms of both utility and running time. I therefore recommend that the paper be accepted. We encourage the authors to make the changes they proposed in their response, especially including high probability error bounds and some discussion of the running time of permute and flip.